# Structural and molecular determinants for the interaction of ExbB from *Serratia marcescens* and HasB, a TonB paralog

Valérie Biou [1,2✉], Ricardo Jorge Diogo Adaixo[3], Mohamed Chami[3], Pierre-Damien Coureux [4], Benoist Laurent[2,5], Véronique Yvette Ntsogo Enguéné [1,2,8], Gisele Cardoso de Amorim [6,9], Nadia Izadi-Pruneyre[6], Christian Malosse[7], Julia Chamot-Rooke [7], Henning Stahlberg[3,10] & Philippe Delepelaire [1,2✉]

ExbB and ExbD are cytoplasmic membrane proteins that associate with TonB to convey the energy of the proton-motive force to outer membrane receptors in Gram-negative bacteria for iron uptake. The opportunistic pathogen *Serratia marcescens* (*Sm*) possesses both TonB and a heme-specific TonB paralog, HasB. ExbB$_{Sm}$ has a long periplasmic extension absent in other bacteria such as *E. coli* (*Ec*). Long ExbB's are found in several genera of Alphaproteobacteria, most often in correlation with a *hasB* gene. We investigated specificity determinants of ExbB$_{Sm}$ and HasB. We determined the cryo-EM structures of ExbB$_{Sm}$ and of the ExbB-ExbD$_{Sm}$ complex from *S. marcescens*. ExbB$_{Sm}$ alone is a stable pentamer, and its complex includes two ExbD monomers. We showed that ExbB$_{Sm}$ extension interacts with HasB and is involved in heme acquisition and we identified key residues in the membrane domain of ExbB$_{Sm}$ and ExbB$_{Ec}$, essential for function and likely involved in the interaction with TonB/HasB. Our results shed light on the class of inner membrane energy machinery formed by ExbB, ExbD and HasB.

[1] Laboratoire de Biologie Physico-Chimique des Protéines Membranaires, Université de Paris, UMR 7099 CNRS, F-75005 Paris, France. [2] Institut de Biologie Physico-Chimique, F-75005 Paris, France. [3] Center for Cellular Imaging and NanoAnalytics, Biozentrum, University of Basel, Mattenstrasse 26, CH-4058 Basel, Switzerland. [4] Laboratoire de Biologie Structurale de la Cellule, BIOC, UMR7654 CNRS/Ecole polytechnique, Palaiseau, France. [5] Plateforme de Bioinformatique, Université de Paris, FRC 550 CNRS, F-75005 Paris, France. [6] Structural Bioinformatics Unit, Department of Structural Biology and Chemistry, C3BI, Institut Pasteur, CNRS UMR3528, CNRS, USR3756 Paris, France. [7] Mass Spectrometry for Biology Unit, CNRS USR 2000, Institut Pasteur, 75015 Paris, France. [8] Present address: Department of Biochemistry, University of Cambridge, Tennis Court Road, Cambridge CB2 1GA, UK. [9] Present address: Instituto de Bioquímica Médica, Universidade Federal do Rio de Janeiro, Rio de Janeiro, RJ, Brasil. [10] Present address: Centre d'imagerie Dubochet UNIL-EPFL-UNIGE & Laboratoire de microscopie électronique biologique UNIL-EPFL, Lausanne, Switzerland. ✉email: valerie.biou@ibpc.fr; philippe.delepelaire@ibpc.fr

Transport of nutrients across the Gram-negative outer membrane is either a diffusion-facilitated or an active process. In the latter case, the process is powered by the energy of the proton-motive force (pmf) transmitted through the periplasm to specialized outer membrane (OM) TonB-dependent transporters (TBDT). A complex of three cytoplasmic membrane proteins, TonB, ExbB, and ExbD, that together form the TonB complex (see ref. [1] for a review) conveys the energy of the pmf to the TBDT. In *Escherichia coli* K12, there is only one set of *tonB, exbB,* and *exbD* genes, whereas there are nine TBDT's[2], all energized by the same complex. ExbB and ExbD respectively belong to the MotA-TolQ-ExbB and MotB-TolR-ExbD protein families. Those proteins are involved in power generation and transmission in various processes (MotAB drives flagellar rotation, the ExbB-ExbD complex referred to as ExbBD energizes active transport of molecules across OM receptors and TolQR is involved in cell division), at the expense of pmf dissipation across the cytoplasmic membrane[3–6]. They form complexes that associate respectively with the flagellar rotor, TonB, and TolA. It has also been shown that ExbD does not accumulate in the absence of ExbB, and that TonB does not accumulate in the absence of the ExbBD complex[7]. The C-terminal domains of TonB and ExbD reside in the periplasm and interact with each other[8,9]. ExbD TM (Trans-Membrane segment) has one strictly conserved aspartate residue[10], which is thought to undergo cycles of protonation/deprotonation (coupled to pmf dissipation) and is essential to its function.

TBDTs comprise a 22-stranded ß-barrel anchored in the outer membrane. An N-terminal domain folded inside the barrel, and called the plug, contains the main region of interaction with TonB, the TonB box, located in the N-terminal periplasm-exposed part of TBDT. The N-terminus of TonB is localized in the cytoplasm, followed by a single transmembrane segment, and by a Pro-Lys rich region long enough to span the periplasmic space and ends with a structured C-terminal domain interacting with the TBDT TonB box. The substrate binds the extracellular side of the TBDT receptor, triggering conformational changes that are transmitted to the periplasmic side and allow the interaction between TonB and the TonB box of the TBDT, leading to the substrate entry into the periplasm by a yet unknown mechanism. A rearrangement of the plug domain has been proposed to occur creating a path for the substrate across the TBDT.

In *Serratia marcescens*, a close relative of *E. coli*, in which about twenty potential TBDTs were identified, there are at least two TonB homologs: an ortholog of $TonB_{Ec}$ (48% amino acid identity with $TonB_{Ec}$[11]) and HasB, a TonB paralog that is strictly dedicated to its cognate outer membrane receptor HasR[12,13]. HasB has the same topology as TonB, and its C-terminal domain interacts specifically with HasR with a circa (ca.) 40-fold higher affinity than the corresponding TonB domain (14). The Has (heme acquisition system) system includes the TBDT HasR, which recognizes both free heme and the high-affinity extracellular heme-binding HasA hemophore. HasB is encoded within the *has* locus and displays low sequence identity with either *E. coli* or *S. marcescens* TonB. The Has system has been functionally reconstituted in *E. coli*. Unlike $TonB_{Sm}$, HasB could not complement $TonB_{Ec}$ functions[12], nor was it able to drive heme entry *via* the HasR receptor in the presence of $ExbBD_{Ec}$. A gain of function mutation in the TM domain of HasB was however isolated in *E. coli*, restoring heme entry *via* HasR in the presence of $ExbBD_{Ec}$[12]. This mutation corresponds to a 6 base-pair duplication leading to a longer TM segment for HasB by inserting AL into CLVLVLALHLLVAALLWP resulting in CLVLVLALALHLLVAALLWP.

Recently several structures of *E. coli* ExbB and ExbBD have been solved, either by X-ray crystallography or cryo-EM[14–16].

All samples studied included ExbB and ExbD, but not all showed an ordered structure of ExbD. In these structures, $ExbB_{Ec}$[14] appeared as a pentamer. Each monomer of ExbB has three TM helices that extend into the cytoplasm, and with a highly polarized charge distribution on the cytoplasmic side. The central pore is apolar, lined by TM helices 2 and 3 of each monomer, creating a large hydrophobic cavity inside the structure. Another study using X-ray crystallography, single-particle cryo-EM and electron diffraction on two-dimensional crystals concluded that ExbB could undergo a pH-dependent pentamer to hexamer transition, and that the hexameric ExbB could accommodate three ExbD TM segments in its pore[15]. A more recent cryo-EM study of ExbBD reconstituted in nanodiscs however confirmed the $ExbB_5ExbD_2$ stoichiometry with two TM helices of ExbD identified in the ExbB pore[16]. This is consistent with previous DEER (Double Electron Electron Resonance) results[14]. Similarly, *Pseudomonas savastanoi* ExbBD exhibits the same stoichiometry[17]. Structures of the flagellar MotAB motor complexes (related to ExbBD) from several bacteria (*Campylobacter jejuni, Bacillus subtilis, Clostridium sporogenes, Vibrio mimicus, Shewanella oneidensis*, PomAB from *Vibrio alginolyticus*) were recently published. They all display a $MotA_5MotB_2$ stoichiometry and share some topology elements with ExbBD complex[17,18].

The discovery of HasB and its functional specificities prompted us to identify the ExbBD complex that would function with it in *S. marcescens*. In this study, we identified the orthologous $ExbBD_{Sm}$ and found that this complex is active with both HasB and TonB. We characterized the family of ExbB proteins with a long N-terminal extension whose presence is strongly correlated to the presence of a *hasB* gene ortholog in the genome. We purified $ExbB_{Sm}$ alone and $ExbBD_{Sm}$ and determined their structures by single-particle cryo-EM at 3.1 and 3.96 Å resolution, respectively. We show that in both cases, $ExbB_{Sm}$ behaves as a stable pentamer; in the $ExbBD_{Sm}$ complex, we observe two TM helices of $ExbD_{Sm}$ in the pore of ExbB pentamer, as was shown for the *E. coli* case. Using NMR measurements, we also show that the N-terminal periplasmic extension specific to $ExbB_{Sm}$ interacts with the C-terminal globular domain of HasB. Finally, *via* mutagenesis studies and bacterial growth assays, we show that the first transmembrane helix TM1 of ExbB contains specificity determinants for interaction with HasB/TonB and that the periplasmic extension of ExbB does play a role in heme acquisition *via* the Has system.

## Results

**Orthologs of *E. coli* *exbB* and *exbD* in *Serratia marcescens* define an ExbB family with an N-terminal extension**. Sequence analysis of strain Db11, a fully sequenced *S. marcescens* isolate (GenBank: *Serratia marcescens* subsp. marcescens Db11, complete genome. ACCESSION HG326223, https://www.ncbi.nlm.nih.gov/nuccore/HG326223.1), identified one putative operon encoding orthologs of *E. coli* ExbB and ExbD (SMDB11_3479(ExbD) and SMDB11_3480(ExbB)). In *E. coli*, the *exbBD* operon is surrounded on the 5' side, and in the opposite direction, by *metC* (encoding a cystathionine lyase) and on the 3' side by *yhgA* (encoding an aldehyde reductase). In *S. marcescens* Db11, the *exbB*-like gene is close to and in the opposite orientation to a *metC* homolog, as in *E. coli*. At the 3' end of the operon, and in the opposite direction there are genes related to sucrose metabolism. A comparison of the coding sequences indicated that the *exbD* gene encoded a 140 residue-long protein (71% identity with the 141 residue-long *E. coli* sequence). In contrast, the putative *exbB* gene encodes a much longer protein than its *E. coli* counterpart (325 residues instead of 244 residues in *E. coli*, 73% identity in the common part). The additional stretch of residues present in $ExbB_{sm}$ is

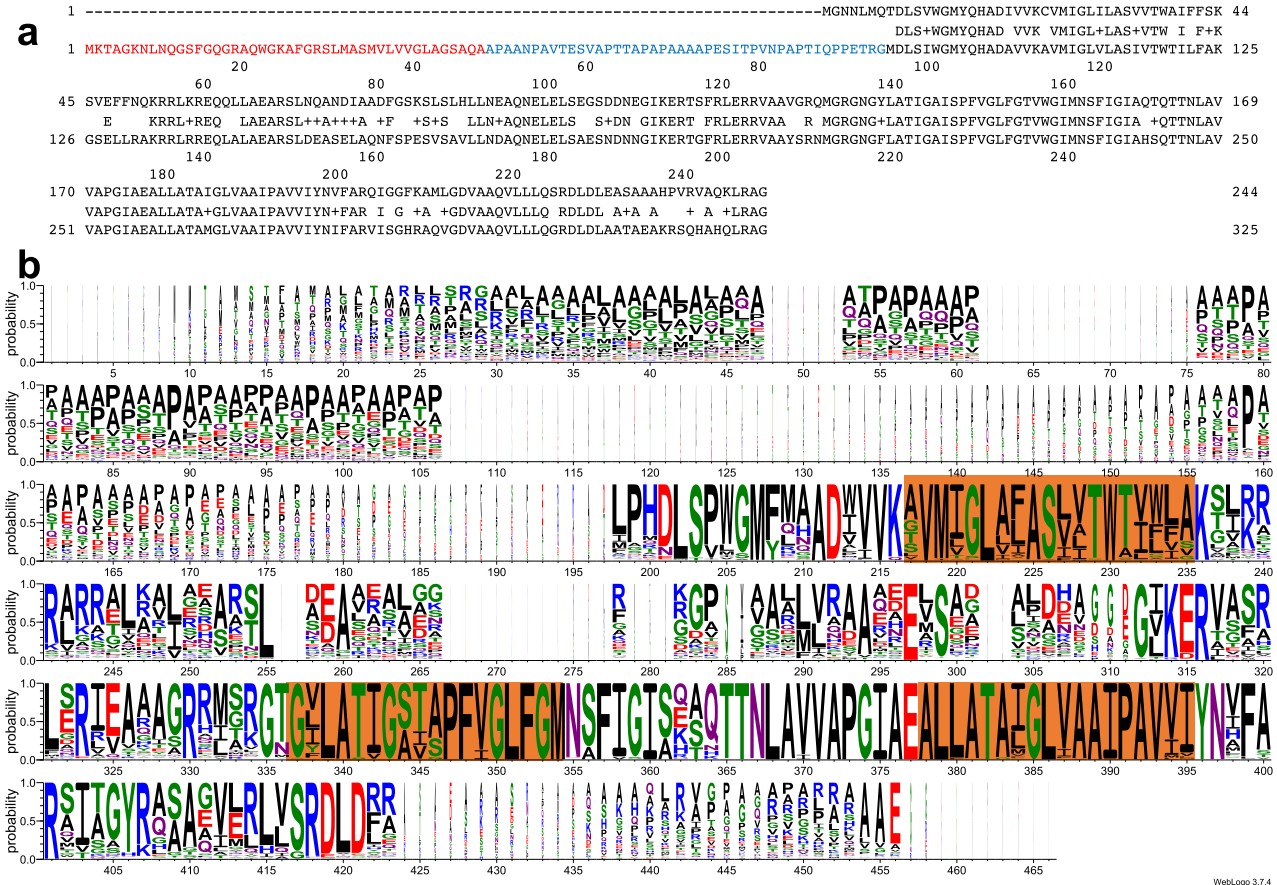

**Fig. 1 ExbB sequence analysis. a** Alignment of ExbB$_{Ec}$ (top) with ExbB$_{Sm}$ (bottom), made with DNA Strider v1.4x-2b, CEA. In red is shown the signal sequence from ExbB$_{Sm}$, and in blue the periplasmic extension, not present in ExbB$_{Ec}$. The consensus sequence is shown on the middle line, where + indicates similar amino acid residues. **b** Weblogo[59] representation of 131 "long" ExbB amino acid sequences aligned with Clustal Omega[60]; those with much shorter periplasmic extensions have been excluded from the alignment. The orange boxes show the position of the three TM segments from the *E. coli* 5SV0 structure.

located at the N-terminus and corresponds to a putative signal peptide followed by a ~50 residue N-terminal extension of the mature protein (Fig. 1a).

A BLAST search[19] on bacterial genomes indicated that such "extralong" ExbB's are found in different Gammaproteobacteria, including *Serratia*, *Yersinia*, *Pseudomonas*, *Erwinia/Dickeya*, and many genera in the Alphaproteobacteria class. Supplementary Table S1 lists representative species of Alphaproteobacteria in which such long ExbB's are found. Interestingly, about 90% of the long ExbB-containing species listed in this table also have a *hasB* gene ortholog.

An alignment of these ExbB amino acid sequences (represented as a Logo on Fig. 1b) also shows that the mature N-terminal extensions are of variable lengths between a few and 150 residues with an average of 50 residues. These extensions are quite rich in Ala (24,6%) and Pro (14,5%) residues and therefore likely to be unstructured but could be involved in protein-protein interactions as proline-rich regions often are in signaling processes[20–22].

The Conserved Domain Database[23] currently contains two ExbB subfamilies in the cI00568 MotA_ExbB superfamily, TIGR02797 (containing *E. coli* ExbB, as well as longer ExbB's) and TIGR02805 (containing *Haemophilus influenzae* ExbB, with a very short cytoplasmic domain between TM1 and TM2). Based on sequence data, TIGR02797 can be divided into 2 groups: those with "extralong" ExbB's in one subfamily, and *E. coli*-like sequences in the other. Along with the existence of HasB in the bacterial species, the identification of this N-terminal addition

prompted us to further characterize the ExbB-ExbD complex from *S. marcescens*.

**ExbBD$_{Sm}$ complements ExbBD$_{Ec}$ for iron acquisition and functionally associates with both TonB and HasB for heme acquisition through HasR.** As a first step in the characterization of the identified ExbBD$_{Sm}$, we tested whether this complex complements ExbBD$_{Ec}$. To this end, the plasmids pBADExbBD$_{Sm}$ and pBADExbBD$_{Ec}$ were constructed, introducing the two genes from *S. marcescens* or *E. coli*, into pBAD24 vector under the control of arabinose-inducible P$_{araBAD}$ promoter. By using growth under iron starvation conditions as a test, we could show that *exbBD$_{Sm}$* complements *exbBD$_{Ec}$*. The *E. coli* C600Δ*exbBD* strain is indeed more sensitive than its wild-type counterpart to iron starvation (induced for example by the Fe$^{2+}$ chelator dipyridyl, DiP) as ExbBD are required to transduce pmf to TonB for siderophore uptake. Its growth in the presence of DiP is restored by the expression of either pBADExbBD$_{Ec}$ or pBADExbBD$_{Sm}$ (Fig. 2).

We then tested the functionality of ExbBDSm in heme acquisition via the Has system reconstituted in a heme auxotroph *E. coli* strain. To avoid interference from the chromosomal *exbBD* operon and from *tonB*, we used the C600Δ*hemA*Δ*exbBD* and the C600Δ*hemA*Δ*exbBD*Δ*tonB* strains, transformed with various recombinant plasmid pairs, one bringing the complete Has locus, with or without the *hasB* gene, the other one with the *exbBD* operon or its derivatives under the control of the P$_{araBAD}$

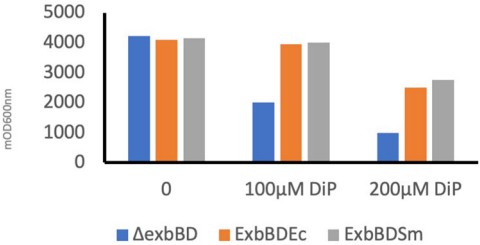

**Fig. 2 ExbBD_Sm complements ExbBD_Ec.** Overnight growth in LB broth of *E. coli* C600*ΔexbBD*(pBAD24) (blue), C600*ΔexbBD*(pBAD24ExbBD_Ec) (orange), and C600*ΔexbBD*(pBAD24ExbBD_Sm) (gray), without (0) or with 100 and 200 µM iron chelator di-pyridyl (DiP). One representative experiment is shown. The vertical axis represents mDO_600nm absorbance units.

promoter. In this kind of experiment, the bacterial growth reflects both the expression of the various <u>Has</u> components under the control of their own regulatory elements[24], and the efficiency of the heme uptake process itself[25]. Two kinds of tests were carried out, one on solid substrate in Petri dishes, allowing to see haloes of bacterial growth around wells punched in the agar and containing the heme source, the other one in liquid medium in microplates with an absorbance plate reader, allowing to record growth at regular intervals over extended periods. The latter type of tests allows a more precise and quantitative description of the phenotypes as it is realized in a homogeneous medium, which is not the case with Petri dishes. Six strains were constructed: one set deleted from TonB C600*ΔhemAΔexbBDΔtonB* (pAMHasISRA-DEB + pBAD24), C600*ΔhemAΔexbBDΔtonB* (pAMHasISRADEB + pBADExbBD_Ec), C600*ΔhemAΔexbBDΔtonB* (pAMHasISRADEB + pBADExbBD_Sm), and one set with TonB C600*ΔhemAΔexbBD* (pAMHasISRADE + pBAD24), C600*ΔhemAΔexbBD* (pAMHasIS-RADE + pBADExbBD_Ec), and C600*ΔhemAΔexbBD* (pAMHasIS-RADE + pBADExbBD_Sm). The ExbBD constructs for Ec and Sm are schematized on the first two lines of Fig. 3a. The results are reported in Fig. 3b (Petri dishes, overnight observation), 3c (Petri dishes, 36 h observation), and 3d (microplates, recording over 60 h). As expected, both in plates and in liquid cultures, control strains (with pBAD24) did not grow (yellow and deep blue in Figure 3b, 3c and 3d). ExbBD_Sm was functional with both HasB and TonB (middle series of holes in Fig. 3b and 3c indicated with orange and blue dots, and orange and blue dots curves in Fig. 3d, Supplementary Data 1), although with quite dramatically different kinetics, the onset of growth occurring at 3-4 h for the HasB-ExbBD_Sm pair, and at 20 h for the TonB-ExbBD_Sm pair. As previously observed, ExbBD_Ec was also functional with TonB (onset of growth at ca. 15 hrs) but not with HasB (bottom series of holes in Fig. 3b and 3c, and green and light-grey dots curves in Fig. 3d). The C600*ΔhemAΔexbBDΔtonB* strain is more sensitive to iron starvation than the C600*ΔhemAΔexbBD* one, as it has no means of acquiring iron (due to the lack of both tonB and exbBD and as hasB does not complement tonB for iron acquisition). A lower dipyridyl concentration is therefore required to achieve comparable iron starvation. This is clear in the central part of Fig. 3b or c, where in the presence of ExbBD_Sm, growth around the wells is observed at 100 µM dipyridyl but hardly at 200 µM in the C600*ΔhemAΔtonBΔexbBD*(pAMhasIS-RADEB) case, the reverse being true in the C600*ΔhemAΔexbBD* (pAMhasISRADE) case (orange vs. pale-blue dots). In liquid cultures in microplates, a lower dipyridyl concentration is required to achieve similar iron starvation as in plates. This is why we used 100 µM for C600*ΔhemAΔexbBD*(pAMHasISRADE) derivatives and 40 µM for C600*ΔhemAΔexbBDΔtonB*(pAMHasISRADEB) derivatives. Therefore, ExbBD_Sm is the *E. coli* ExbBD ortholog, able to associate both with HasB and TonB_Ec.

**Characterization of ExbB_Sm and ExbBD_Sm: Specific function of the N-terminal extension of ExbB_Sm.** To gain further insight into the possible differences between ExbBD_Sm and ExbBD_Ec, we purified both ExbB_Sm and ExbBD_Sm (as C-terminal 6-His-tagged proteins on ExbB for ExbB purification, and C-terminal 6-His tag on ExbD for ExbBD purification, see Materials and Methods section) in LMNG (lauryl maltose neopentyl glycol) micelles (Supplementary Fig. S1a, b). Both ExbB_Sm and ExbBD_Sm appear as homogeneous oligomers at the last size exclusion chromatography (SEC) step.

Mass spectrometry analysis showed that purified ExbD_SmHis6 has a mass of 16,131 Da close to its theoretical mass of 16261.7 Da. Purified ExbB_Sm has a measured mass of 29,557 Da and its predicted mature sequence has a theoretical mass of 29,556.8 Da. Together with the determination of the first amino acid residues for ExbB, this confirmed that the predicted signal sequence of ExbB_Sm is absent from the mature sequence and has therefore been processed. It also showed that ExbD initial Met residue was cleaved off (see Supplementary Fig. S2). More interestingly, mass spectrometry analysis of chloroform/methanol extracts of the purified proteins also showed that both complexes contained native lipids (Supplementary Fig. S3), mostly phosphatidylglycerol (PG) and phosphatidylethanolamine (PE), and that ExbBD also contained some cardiolipin (CL) (Supplementary Fig. S4). Further analysis of aliphatic chain composition of lipids shows evidence of specific composition with a majority of PG with 34 carbons and 2 unsaturations (Supplementary Fig. S5). Considering that the ExbBD_Sm complex is functional in *E. coli*, and that the phospholipid composition of *S. marcescens* is quite similar to that of *E. coli*[26], this affinity for lipids may disclose a specific recognition that may be important for the function.

The presence in ExbB_Sm of an N-terminal extension residing in the periplasm led us to investigate its interaction with HasB_CTD by NMR. This extension contains more than 50% of Ala and Pro residues and is predicted to be unstructured by disorder prediction servers such as IUPRED2[27]. Since this region is predicted to be in the periplasm where the C-terminal domain of HasB, HasB_CTD is also located, we investigated the potential interaction of the ExbB_Sm 1–44 synthetic peptide with HasB_CTD by NMR. The analysis of chemical shift perturbations (CSP) of amide resonances of HasB_CTD upon addition of the peptide, showed that the chemical environment of their corresponding residues was modified because of their interaction (Fig. 4a and Supplementary Fig. S6). Perturbed residues are mainly located on the helical face of HasB_CTD forming a continuous surface of interaction (Fig. 4b, c) (R175, R178-K180, K184, Q192, T200, L201, Q204, H206, A232, A240, G246). Interestingly, this face is on the opposite side of the third beta strand of HasB that was previously shown to interact with HasR[28]. In addition, the residues of a small pocket at the C-terminus of HasB (D255, R259) show also high CSP and might be involved in the interaction with ExbB or subject to a conformation change induced by this interaction.

To assess the potential function of the N-terminal periplasmic extension of ExbB_Sm, we engineered a ExbB_Smdelextss mutant lacking this extension (Fig. 3a) (see "Materials and Methods" for details) and tested its effect on the growth of a C600*ΔhemAΔexbBDΔtonB* strain harboring plasmids pAMHasISRADEB and pBADexbBD_Sm or its derivatives in liquid culture. We observed that the lag between the start of the experiment and the onset of growth is at least 2 h longer for the ExbB_Smdelextss mutant strain, as compared to the WT strain (Fig. 5a, compare orange and yellow curves) and a control strain (medium blue). To ensure that this difference was not due to a difference in expression of the two constructs, we compared their amounts in membrane preparations by immunodetection. As our anti-

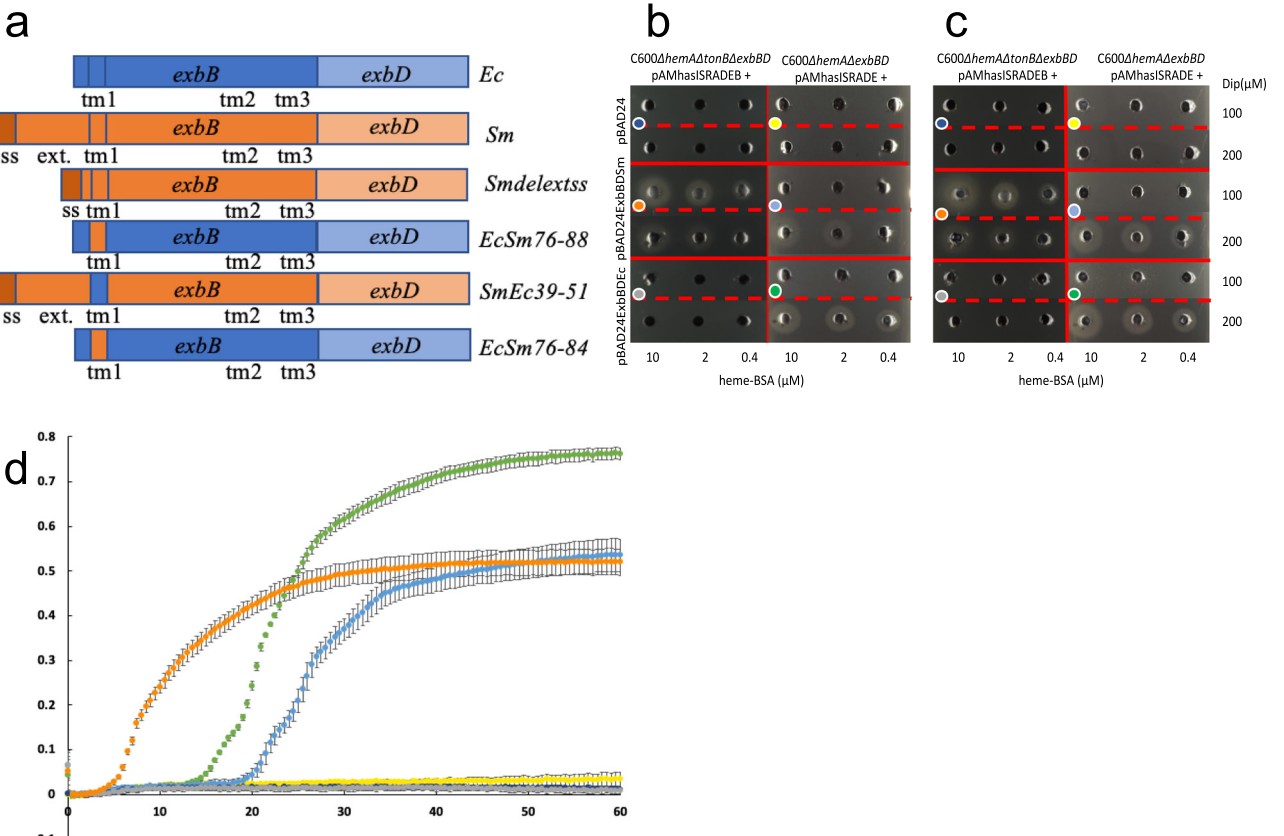

**Fig. 3 Role of HasB and TonB on bacterial growth. a** Representation of the constructions used in **b**–**d**, and elsewhere in this work. ss refers to the signal sequence, ext to the periplasmic extension of ExbB$_{Sm}$, and tm1, tm2, and tm3 to the first, second, and third ExbB transmembrane segments, respectively. **b**, **c** Growth around wells of *E. coli* C600Δ*hemA*Δ*tonB*Δ*exbBD* harboring pAMHasISRADEB and specified recombinant plasmids (**b**, **c** left parts) or *E. coli* C600Δ*hemA*Δ*exbBD* harboring pAMHasISRADE and specified recombinant plasmids (**b**, **c** right parts); the pictures were taken after overnight growth at 37 °C (**b**), or after 36 h growth (**c**). Arabinose concentration was 40 μg/ml. Dipyridyl (DiP) iron chelator concentration was either 100 or 200 μM. Heme-BSA concentrations inside the wells were 10, 2, and 0,4 μM. **d** growth curves in microplates of the same strains as in **b**, **c**; DiP concentration was 100 μM in the C600Δ*hemA*Δ*exbBD*(pAMHasISRADE) derivatives, and 40 μM for C600Δ*hemA*Δ*tonB*Δ*exbBD*(pAMHasISRADEB) derivatives, arabinose concentration 4 μg/ml, Heme-BSA 1 μM. The error bars correspond to standard error calculated from the duplicate measurements. Colored dots in (**b**, **c**) refer to the equivalent growth conditions in (**d**).

ExbB$_{Sm}$ antibody was not sensitive enough for this measurement, we used the His-tagged version of ExbB$_{Sm}$ readily detectable with anti-His6 antibodies. Coomassie blue-stained SDS-PAGE showed that the 3 conditions have similar total amounts of protein (Fig. 5b, Supplementary Fig. S7a). The Western blot shows that the amount of the ExbB$_{SmHis6}$ variant deleted from its N-terminal extension is at least equal, if not slightly higher, to the amount of the wild-type protein (Fig. 5c, lanes 2 and 3, Supplementary Fig. S7b), ruling out a possible decrease in protein concentration. The N-terminal periplasmic extension of ExbB$_{Sm}$, is therefore likely involved in the functioning of the Has system, whether at the transcription activation of the Has locus, and/or at any later step.

**ExbB$_{Sm}$ and ExbBD$_{Sm}$ structural analysis by cryo-EM show ExbB$_5$ and ExbB$_5$D$_2$ stoichiometries**. As ExbB$_{Sm}$ presented specific features compared to ExbB$_{Ec}$, we set out to determine its structure by single-particle cryo-EM, see Material and Methods for details. The 3D class average model clearly showed a pentameric structure. It was refined and polished to obtain a resolution of 3.1 Å using the Fourier shell correlation (FSC) gold-standard criterion at 0.143. Maps were refined with and without

C5 symmetry and showed a 98% correlation in density. Therefore, we chose to use the map based on C5 symmetry for model building.

The structure solved here and shown in Fig. 6a (side view) and b (cytoplasm view), has the same α-helix bundle topology and is very similar to that of *E. coli* ExbB 5SV0 structure (that was co-purified with ExbD but did not show any ordered density for ExbD (14)), with a 1.3 Å root mean square deviation (rmsd) over all Cα atoms (Table 1). This shows that ExbB$_{Sm}$ is stable as a pentamer on its own. The periplasmic N-terminal extension did not yield any visible density, precluding its structure determination. At the C-terminus of each monomer, however, density was clearly defined, allowing structure determination for an additional 10 residues in a helical conformation up to the last one (helix α8 finishing with Gly 283, see Fig. 6c) before the His-tag that is present but disordered. In the 5SV0 X-ray structure, a calcium ion, present in the crystallization solution, is bound to the five Glu 106 (of helix α2 in the TM region) on the cytoplasmic side. In ExbB$_{Sm}$, this residue is replaced by an Asn, and we do not observe any density that could be assigned to a metal ion.

The structure of the whole pentamer with one monomer colored as a function of sequence conservation in this sub-class of

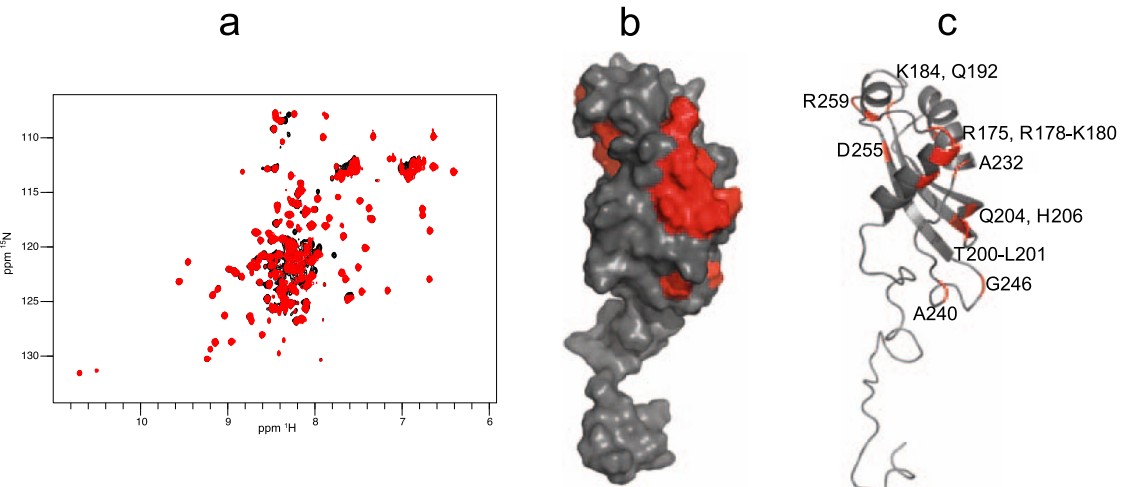

**Fig. 4 Interaction of HasB$_{CTD}$ with the periplasmic fragment of ExbB$_{Sm}$, as detected by NMR. a** Superposed $^1$H–$^{15}$N HSQC spectra of 0.15 mM $^{15}$N-labeled HasB$_{CTD}$ in 50 mM sodium phosphate, pH 7, 50 mM NaCl in the presence (red) or absence (black) of the periplasmic peptide of ExbB$_{Sm}$ residues 1–44. **b, c** HasB$_{CTD}$ residues (PDB code 2M2K) exhibiting the highest CSP in the presence of the periplasmic peptide of ExbB$_{sm}$ are colored red (**b**: surface representation; **c**: cartoon representation). The residues showing the highest CSP are indicated.

ExbB (Supplementary Fig. S8) shows that the highest residue conservation is inside the transmembrane channel, indicative of high functional constraints, while the TM residues located at the membrane surface are more variable. The same observation was made for ExbB sequences lacking N-terminal signal sequence and periplasmic extension[14].

The cryo-EM density of ExbB$_{Sm}$ shows vents located at the interface of adjacent monomers and at the height of the cytoplasmic junction with the inner membrane leaflet (Fig. 6a). These vents may allow solvent or ion passage. Additional density was also clearly observed on the external surface, and we attributed it to the phospholipids present in the preparation. Each ExbB monomer appears to be associated with the equivalent of two PG molecules, located at the inner leaflet of the cytoplasmic membrane (Fig. 6a, d, e). However, we were able to model only one PG molecule per monomer with confidence. Three PE (phosphatidylethanolamine, the major *E. coli* phospholipid) molecules and one PG molecule were identified as associated to the ExbBD pentamer in the *E. coli* complex after reconstitution in nanodiscs[16]. The cryoEM density map, when displayed at a level that shows the detergent belt, shows density inside the ExbB membrane pore (Supplementary Fig. S9a). This density is too noisy to allow model building (Supplementary Fig. S9b). However, it corresponds to a region with positive electrostatic charge on the top and bottom of the pore and neutral or hydrophobic in the middle (Supplementary Fig. S9c). This density could be due to the presence of lipid or detergent. Interestingly, it is located at a different height as compared to the detergent belt and external lipids. In summary, our cryo-EM structure shows that ExbB$_{Sm}$ is stable as a pentamer and associates strongly with specific phospholipids coming from the inner leaflet of the membrane.

We also solved the structure of the ExbBD$_{Sm}$ complex by cryo-EM (Fig. 7 and Table 1). As for ExbB, the purified complex exhibited a symmetric peak on SEC (see Supplementary Fig. S1b). Cryo-EM data were collected and processed as described in the Material and Methods section. In contrast to ExbB$_{Sm}$, 2D classes showed preferential orientations with 92% top views and only 8% side views. Due to this strong bias in particle orientation, the resolution achieved was 3.96 Å, precluding reliable positioning of the side chains. However, our ExbB$_{Sm}$ model fitted readily into the density. As observed for ExbB$_{Sm}$ alone, in the ExbBD$_{Sm}$

complex, ExbB$_{Sm}$ behaves as a pentamer, and no sign of other assemblies was found. Similarly, no density could be attributed to the periplasmic extension. Two clear densities inside the ExbB$_{Sm}$ pore were assigned to the pair of ExbD$_{Sm}$ TM segments (Fig. 7b, c). The charge distribution is highly polarized on the cytoplasmic side: the region close to the membrane is positively charged and the distal part is negatively charged as observed for ExbB$_{Ec}$ (Supplementary Fig. S10a). The central pore of ExbB$_{Sm}$ is apolar, lined by TM helices 2 and 3 of each monomer, creating a large hydrophobic cavity inside the structure (Supplementary Fig. S10b). The ExbD helices were initially oriented similarly to ExbD$_{Ec}$ in structure 6TYI and remained stable during refinement. Interestingly, the ExbD TM helices are at the same height as the density observed in the pore of ExbB alone, and closer to the periplasm than the membrane bilayer. In the refined model, Asp 25 from ExbD$_{Sm}$ monomer chain G faces Thr 218 from ExbB$_{Sm}$ chain C, while Asp 25 from the ExbD$_{Sm}$ chain F faces the interface between two ExbB$_{Sm}$ monomers A and E (Supplementary Fig. S10c, d). The estimation of pKa, by using the program Propka[29] server, shows that Asp 25 has a pKa of 7.3 for chain F and 7.4 for chain G, both very peculiar for solvent-accessible acidic residues but that can be found in buried active sites or membrane proteins[30]. This pKa should allow protonation and deprotonation of ExbD Asp 25 at physiological pH.

As compared to the structure of ExbB$_{Sm}$, there is a slight «opening» towards the periplasmic side (the distance of Ala 197 from one subunit to Leu 204 of the facing subunit varying from 25.5 Å in ExbB$_{Sm}$ to 29.8 Å in ExbBD$_{Sm}$, much greater than any error expected due to the difference in resolution of the two structures). This opening of the structure is limited to the periplasmic part (Supplementary Fig. S11). The rmsd between ExbB$_{Sm}$ alone and in complex with ExbD$_{Sm}$ is 2 Å over all Cα of the pentamer, and the rmsd between ExbB$_{Sm}$ in complex with ExbD$_{Sm}$ and ExbB$_{Ec}$ in complex with ExbD$_{Ec}$ (structure 6TYI chains A-E) is 1.7 Å.

As compared to the 6TYI structure of *E. coli* ExbBD complex, we observe significant differences that may be related to the specificity of the function of ExbB$_{Sm}$. Both ExbB$_{Sm}$ and ExbBD$_{Sm}$ inner pores are slightly wider than their *E. coli* counterparts at the periplasmic entrance. Consequently, two channels cross the membrane region, (extending from the periplasmic entrance to

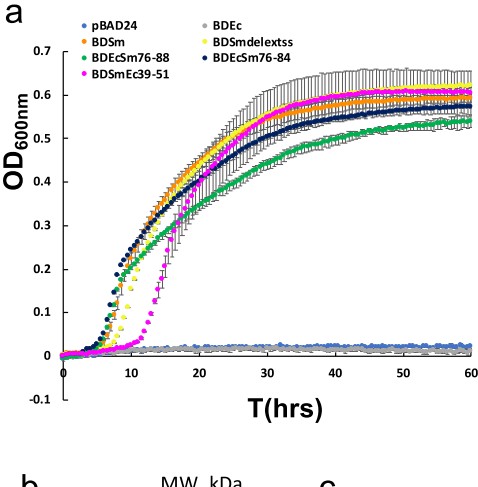

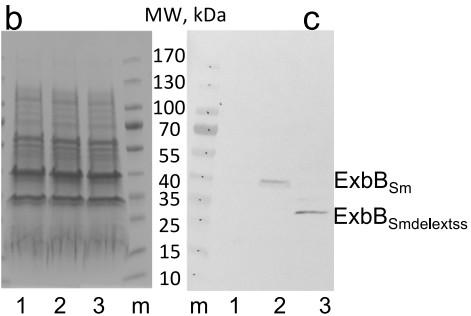

**Fig. 5 ExbB TM1 interacts with HasB. a** bacterial growth curves of *E. coli* C600Δ*hemA*Δ*exbBD*Δ*tonB* harboring pAMHasISRADEB plasmid together with either pBAD24 (medium blue), pBADExbBD$_{Sm}$ (orange) or pBADExbBD$_{Smdelextss}$ (yellow), pBADExbBD$_{Ec}$ (gray), pBADExbBD$_{EcSm76-88}$ (green), pBADExbBD$_{EcSm76-84}$ (dark-blue), pBADExbBD$_{SmEc39-51}$ (magenta) in the presence of 1 μM He-BSA as heme source, 4 μg/ml arabinose to induce ExbBD expression and 40 μM dipyridyl to induce iron starvation (see Materials and Methods for further details). The optical path was ca. 3 mm long, and each curve represents the mean of four replicates of the same culture, recorded every 30 min for 66 h. The error bars are calculated as the standard error of the four measurements. Coomassie-blue stained gel (**b**) and immunodetection (**c**) with anti-His6 antibody of membrane preparations of *E. coli* C600 harboring either pBAD24 plasmid (1), pBADExbB$_{Sm}$ (2) or pBADExbB$_{Smdelextss}$ (3). The equivalent of 0.6 OD$_{600nm}$ was loaded in (**b**), and of 0.3 OD$_{600nm}$ in (**c**).

the Asp 25 residue from the ExbD TM deeply embedded in the ExbB pore) that are clearly seen in the ExbBD$_{Sm}$ (as detected by the MoleOnline[31] Server, Supplementary Fig. S12a, c; the average diameter is around 3 Å) while the *E. coli* structure exhibits a single, much thinner channel as detected with the same parameters of MoleOnline (Supplementary Fig. S13b, d; average diameter around 2 Å). It is therefore possible that this structure represents another state of the ExbBD motor complex, whereby the periplasmic side allows solvent access to the Asp 25 residue of ExbD TM. The different physico-chemical conditions (nanodiscs vs. detergent micelles, 200 vs. 100 mM NaCl, pH7,4 vs. pH8) might also influence such parameters.

In summary, we show that ExbBD from *S. marcescens* has a 5:2 stoichiometry with a larger channel allowing solvent or proton transport from the periplasmic side as compared with the *E. coli* structure.

**Swapping residues in TM1 between ExbB$_{Sm}$ and ExbB$_{Ec}$ strongly suggests that ExbB TM1 interacts with HasB/TonB.** As already mentioned, TBDT function requires a productive

association between TonB or its orthologs/paralogs with ExbBD complexes. It is known that the periplasmic domains of ExbD and TonB interact, and there are both genetic and structural pieces of evidence for the interaction between the TM helix of TonB and the first TM helix of ExbB. Our experimental system gave us the opportunity to investigate the specificity determinants between HasB/TonB and ExbB. The previous work[12] lends support to the absence of interaction between HasB and ExbBD$_{Ec}$. However, our data rather favor a non-functional interaction between HasB and ExbBD$_{Ec}$. As it is known that ExbBD is required to stabilize TonB, we first tested whether HasB was stabilized by ExbBD. We could show that, although HasB does not accumulate in the absence of ExbBD, it is readily present in comparable amounts with either ExbBD$_{Sm}$ or ExbBD$_{Ec}$ (Fig. 8 and Supplementary Fig. S13). This observation therefore indicates that most likely HasB also interacts with ExbBD$_{Ec}$ so as to be stabilized (i.e., withstanding proteolytic degradation), but in a non-functional manner.

A general framework put forward for the Mot complexes indicates that MotA (equivalent of ExbB) rotation around MotB (equivalent of ExbD) drives the rotation of the flagellum basal ring[17,18]. We tested the possibility that ExbB residues exposed at the surface of the protein might be involved in a functional interaction with HasB/TonB, as has been proposed in the case of the TonB-ExbBD complex from *P. savastanoi*[17]. The superposition of residue conservation on the structure of ExbB showed that the membrane-facing residues of the TM region of helix α2 and the cytoplasmic residues were the least conserved (Fig. 1 and Fig. S8). We noted one conspicuous stretch of residues located in the cytoplasmic leaflet of TM helix 1 (see Supplementary Fig. S8):

```
residues    76-88    of ExbB_Sm  TILFAKGSELLRA
corresponding to
residues 39-51 of ExbB_Ec AIFFSKSVEFFNQ
```

(identical residues are underlined) that is the least conserved one between the two proteins. In this stretch, the conserved I, K and E residues (Fig. 9a–d) and point toward the innermost part of the protein and the non-conserved residues point to the outside and are available for protein-protein interaction as shown by a helical wheel representation (Fig. 9e, f).

We speculated that this region of ExbB might establish interaction with HasB. We made a chimeric ExbB$_{Ec}$ protein, with its 39–51 region replaced by residues 76–88 from ExbB$_{Sm}$, named ExbB$_{Ec-Sm76-88}$, and asked how this chimeric ExbB$_{Ec}$ protein would allow growth in the presence of HasB (see Fig. 3a for the constructions), as compared to ExbB$_{Ec}$. In this last case, as already seen, there is no growth in the presence of HasB and ExbBDEc (Fig. 5a, grey curve), whereas the growth in the presence of ExbB$_{Ec-Sm76-88}$ and HasB is indistinguishable (or even with a quicker onset) from the one with ExbBDSm and HasB (Fig. 5a, green curve, compared to orange curve). To better locate the region responsible for specificity, we produced one additional mutant where only residues 76–84 (TILFAKGSE) from ExbB$_{Sm}$ were exchanged for residues 39–47 (AIFFSKSVE) from ExbB$_{Ec}$ yielding ExbB$_{Ec-Sm76-84}$ (see Fig. 3a). ExbB$_{Ec-Sm76-84}$ was quite comparable to ExbB$_{Ec-Sm76-88}$ (see deep-blue and green curves in Fig. 5a). This 9-residue stretch is therefore sufficient to alter ExbB$_{Ec}$ to be better adapted to HasB. This set of experiments shows that the intramembrane functional zones of ExbB are crucial for the growth of *E. coli* in our experimental setup and therefore likely govern the interaction between ExbB and HasB. The substitution, in ExbB$_{Sm}$ TM1 helix of the corresponding residues from ExbB$_{Ec}$ (ExbBD$_{SmEc39-51}$), degraded the activity of ExbB$_{Sm}$, with a longer lag before onset of growth (10 vs. 3-4 hrs, orange and magenta curves).

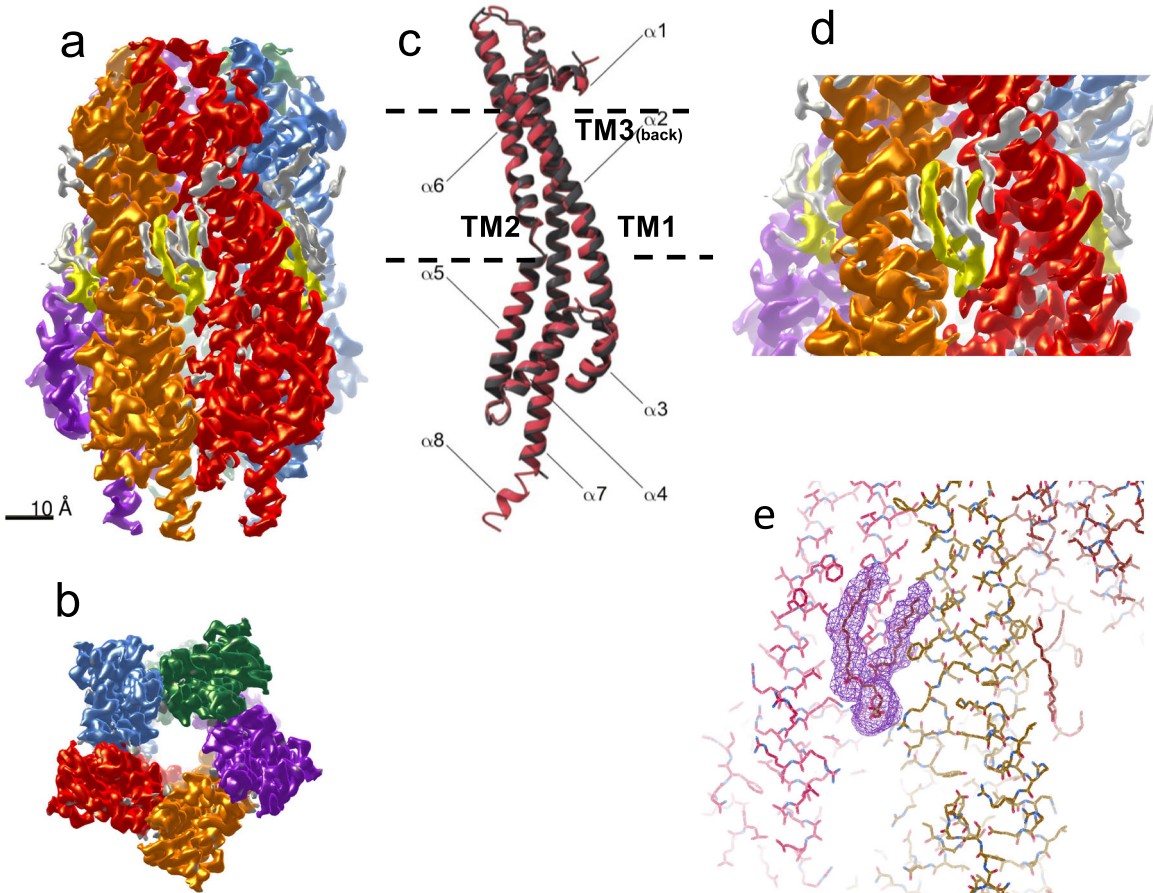

**Fig. 6 Pentameric structure of ExbB$_{Sm}$ solved by cryo-EM. a** the cryo-EM density map is shown, each monomer with distinct colors, view from the side. The yellow/gray regions represent non-protein density. The yellow regions were modeled as PG molecules. **b** View from the cytoplasmic side. **c** Superposition of a monomer of ExbB$_{Ec}$ (dark gray) and ExbB$_{Sm}$ (red), represented as ribbon. The membrane thickness is represented by the dotted lines and the transmembrane segments TM1 (part of α2 helix), TM2 (part of α6 helix) and TM3 (part of α7 helix) are indicated. **d** close-up representation of one PG density; **e** fitting of the PG molecule inside the density; nearby ExbB residues are depicted as sticks using the Coot software[49].

In summary, we showed using NMR and culture growth assays that ExbB$_{Sm}$ N-terminal extension (section 3b) and a few residues in the α2 helix TM region (section 5) are likely involved in specific interactions with HasB.

## Discussion

**ExbBD$_{Sm}$ is a 5:2 complex at pH 8.** Our structures were solved at pH 8, and examination of the 2D class averages only showed pentameric structures and not hexameric ones, contrarily to what has been observed in the Maki-Yonekura study, where ExbB was mostly seen as a hexamer in the high pH regime[15]. In their study, the solubilizing detergent was DDM, that was subsequently exchanged for either C8E4 or C10E5. These shorter chain detergents might have led to some destabilization of the pentameric structure and provoked the conversion to the hexameric form. Our data agree with those published by Lloubes, Buchanan et al.[14,16]. In vivo, a stoichiometry of 7ExbB, 2ExbD and 1TonB has been determined in *E. coli*[32]. Although one cannot completely rule out some bias in this indirect measurement, a physiologically relevant reason for an excess of ExbB would be to provide a permanent scaffold to which TonB and/or ExbD could associate with, upon TonB complex dissociation, that might occur during the catalytic cycle, as has already been proposed[33].

**ExbB and ExbBD are co-purified with endogenous lipids.** In their last study, Celia *et al.* reconstituted ExbBD from *E. coli* into

nanodiscs with added phospholipids from *E. coli*, and found mostly PE bound to ExbBD[16]. It is unclear whether this phospholipid was a genuine tightly bound, co-purified lipid or whether it came from the lipids added during reconstitution in the nanodiscs. In our case, we did not add any lipid and our mass spectrometry identification of co-purified lipids found mostly PG and some PE in the ExbB case (and CL for ExbBD). The cryo-EM structure shows that they are located in the inner leaflet, as in the *E. coli* complex. PG and CL are negatively charged, whereas PE is zwitterionic. This might be pertaining to the functioning of the complex, as the lipids are bound to a highly positively charged interface, very close to the fenestrations seen in the structure and PE might therefore be less tightly bound than PG or CL. Further analysis by mass spectrometry of the aliphatic chains of the lipids present in our sample also revealed a specific composition in their length and unsaturated nature.

One point also worth mentioning is the existence of "channels" inside the ExbBD tunnel, potentially allowing the passage of proton/hydronium ion up to the Asp 25 residue of ExbD, deeply embedded in the apolar medium of ExbB, and therefore with a pKa close to physiological pH, allowing easy protonation/deprotonation steps of Asp 25 side chain. The identified channels connect the cytoplasm to the periplasm *via* Asp 25, allowing us to propose a possible trajectory for the proton transfer via those channels.

**Motor model proposal.** Regarding the coupling of the pmf dissipation with the mechanical work carried out by MotAB/TolQR/ExbBD complexes, several models have been put forward that

**Table 1 Data collection, processing, and refinement statistics for ExbB$_{Sm}$ and ExbBD$_{Sm}$.**

|  | ExbB$_{Sm}$ (EMDB-10789) (PDB 6YE4) | ExbB$_{Sm}$-ExbD$_{Sm}$ (EMDB-11806) (PDB 7AJQ) |
|---|---|---|
| *Data collection and processing* |  |  |
| Magnification | 165,000 | 139,000 |
| Voltage (kV) | 300 | 300 |
| Electron exposure (e−/Å$^2$) | 56 | 55 |
| Defocus range (μm) | −1.5 to −2.5 | −1 to −3 |
| Pixel size (Å) | 0.87 | 1.067 |
| Symmetry imposed | C5 | C1 |
| Initial particle images (no.) | 850,000 | 1,373,000 |
| Final particle images (no.) | 157,000 | 158,000 |
| Map resolution (Å) | 3.1 | 3.96 |
| FSC threshold | 0.143 | 0.143 |
| Map resolution range (Å) | 3.1–5.5 | 3.96–8 |
| *Refinement* |  |  |
| Initial model used (PDB code) | Phyre2 homology model based on 5SV0 | 6YE4 |
| Model resolution (Å) | 3.1 | 3.96 |
| FSC threshold | 0.143 | 0.143 |
| Model resolution range (Å) | 2.8–9 | 3.5–11 |
| Map sharpening *B* factor (Å$^2$) | 200 | 200 |
| Model composition |  |  |
| Non-hydrogen atoms | 9145 | 9256 |
| Protein residues | 1180 | 1229 |
| Ligands | PGT: 5 | 0 |
| *B* factors (Å$^2$) |  |  |
| Protein min/max/mean | 21.58/ 145.06/ 45.97 | 93.07/ 411.58/ 169.35 |
| Ligand min/max/mean | 45.43/ 46.91/ 46.12 |  |
| R.m.s. deviations |  |  |
| Bond lengths (Å) | 0.026 | 0.003 |
| Bond angles (°) | 1.56 | 0.573 |
| Validation |  |  |
| MolProbity score | 1.1 | 2.03 |
| Clashscore | 3 | 15.3 |
| Poor rotamers (%) | 0 | 0 |
| Ramachandran plot |  |  |
| Favored (%) | 99 | 95 |
| Allowed (%) | 1 | 5 |
| Disallowed (%) | 0 | 0 |

might be more or less easy to reconcile with the structural data. A detailed mechanism was proposed for *C. jejuni* MotAB complex, where cycles of protonation/deprotonation of the conserved Asp of the MotB TM (equivalent to the conserved Asp 25 of the ExbD TM) are coupled to the rotation in discrete steps of the MotA pentamer around MotB axis, provided that the periplasmic domain of MotB is anchored to the peptidoglycan layer *via* its peptidoglycan binding site[18]. Our ExbBD structure indeed shows that ExbD monomers have two orientations relative to ExbB: Asp 25 from one monomer faces Thr 208 of ExbB and Asp 25 from the second monomer faces a hydrophobic region at the interface of two ExbB monomers. In the first state previously protonated Asp25 could be deprotonated in contact with polar Thr208 while the second, deprotonated Asp 25, could pick a proton from the periplasm via the channel. Rotation of the ExbB pentamer around the two ExbD chains would then lead to a new protonation/deprotonation cycles accompanied by a change in the environment upon rotation. How is this rotative energy conveyed to the TBDT?

The binding of an iron-loaded substrate on the extracellular binding site of a TBDT triggers a reaction cascade ultimately leading to the substrate entry into the periplasm. The inner membrane TonB(HasB) complex conveys the energy of the pmf to the TBDT most likely by a rearrangement of the plug inside the barrel to allow substrate access to the periplasm and its capture by periplasmic binding proteins. Specific interactions between the TonB(HasB) box of the receptor and the C-terminal domain of TonB(HasB) are essential in this process[28]. A wealth of data has

also accumulated documenting specific interactions between TonB$_{CTD}$ and ExbD$_{CTD}$, depending upon the energy state of the cell. It is also known that ExbD and TonB interact *via* their periplasmic parts[34] and in particular residue 150 of TonB (positioned just upstream the C-terminal globular domain) can make cross-links to the C-terminal domain of ExbD[35]. There have also been indications and suggestions that both TonB$_{CTD}$ and ExbD$_{CTD}$ could interact with the peptidoglycan sacculus, providing anchor points to allow force transmission. Molecular modeling works hypothesized that the C-terminal domain of TonB[36] and the C-terminal domain of ExbD[37] have specific binding sites for the peptidoglycan network. Atomic force microscopy experiments also showed that by exerting a pulling force on the C-terminal domain of TonB bound to the BtuB TBDT, a partial unfolding of the TBDT plug occurs, potentially leading to the entry of the substrate in the periplasm[38]. Our NMR results also show that ExbB interacts *via* its alanine and proline-rich periplasmic N-terminal region with the HasB periplasmic domain on the side opposite to that of the TonB box interaction, thus rendering a tripartite interaction possible. Our in vivo growth results with the ExbB$_{Smdelextss}$ mutant devoid of the periplasmic extension point to a possible role of the N-terminal periplasmic extension in the activation of the transcription of the Has locus (Fig. 5).

The MotA-MotB model also posits that MotA outer region interacts with the rotor of the flagellum and thus that MotA rotation drives the rotation of the flagellum. Similarly, the MotAB rotating model can be extrapolated to the ExbBD complex. In this model, ExbB would rotate around ExbD, driving the rotation of TonB/HasB thanks to the specific interaction between TonB/HasB TM and ExbB TM α2. This kind of interaction is supported by our mutagenesis data, as we could strongly increase the efficiency of ExbB$_{Ec}$ with HasB by exchanging a short stretch of residues between ExbB$_{Sm}$ and ExbB$_{Ec}$. It is also in line with a previous TonB TM point mutant partially suppressed by an ExbB α2 point mutant[39]. Structure comparison of the swapped regions shows that ExbB$_{Sm}$ has smaller residues that may be better accommodated and less specific than the *E. coli* sequence (Fig. 9a, b, compare structures in Figs. 9c, d and helical wheels on Figs. 9e, f). Similar interactions have been proposed by Deme et al. on the TonBExbBD complex from *P. savastanoi*[17], where an extra cryo-EM density is seen outside the ExbB pentamer and was tentatively assigned to the TM domain of TonB. The residues from ExbB$_{Ec}$TM1 (S34 and A39) identified as co-evolving with TonB TM region (18) are only a little deeper in the bilayer than the residues we exchanged between ExbB$_{Ec}$TM1 (39-51) and ExbB$_{Sm}$TM1, that are closer to the cytoplasm. More specifically, whereas S34 is conserved between ExbB$_{Ec}$ and ExbB$_{Sm}$ (S71), A39 is replaced by T76. Moreover, the hydrophobic core of HasB TM is likely shorter than that of TonB as seen in the sequence alignment of HasB and TonB TM domains:

```
HasBSmTM 18-39 RRC---LVLVLALH-LLVAALLWPRR
TonBSmTM 10-34 RRISVPFVLSVGLHSALVAGLLYAS-
TonBEcTM 07-31 RRFPWPTLLSVCIHGAVVAGLLYTS-
consensus     RR    L  LH  VA LL
```

This difference in length might influence the orientation of HasBTM inside the bilayer and potentially explain the gain of function of HasB6 mutant (in which the hydrophobic core has two more residues (39)) to be better suited to ExbB$_{Ec}$ than HasB. The presence of a proline residue in the TonB TM may also change its shape and the interaction with ExbB.

Several models have been proposed to account for the functioning of the TonB complex, in conjunction with the entry

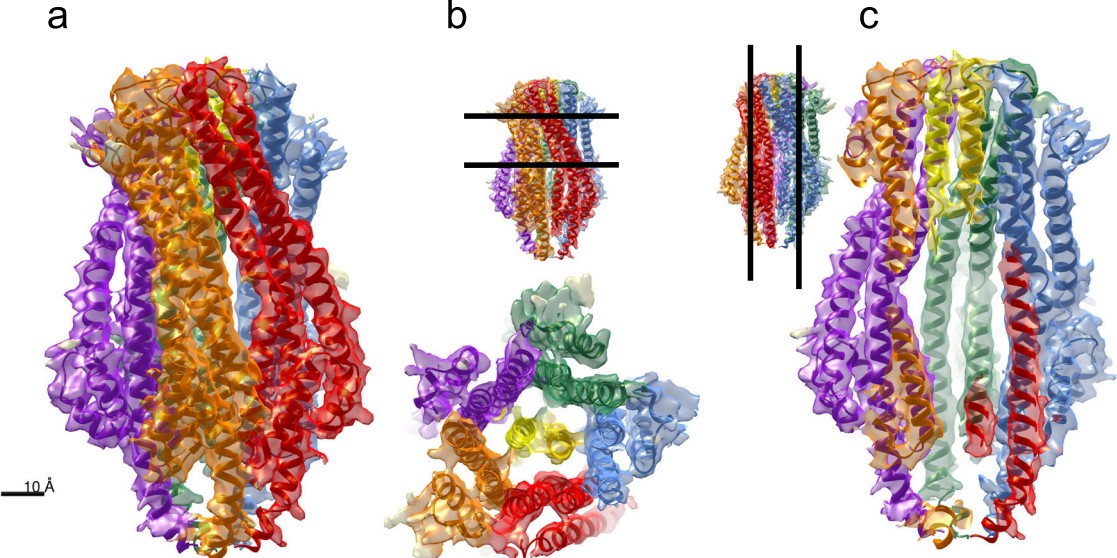

**Fig. 7 Structure of ExbBD_Sm solved by cryo-EM.** The same color code as in Fig. 7 for is used the ExbB monomers and the two ExbD monomers are colored yellow and gold. **a** Side view. **b** View from the periplasmic space. **c** Clipped side view, with the helix of each ExbD monomer represented (yellow and gold). For (**b**, **c**), the insets show the clipping planes.

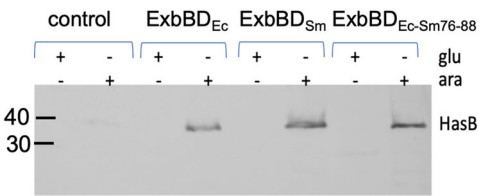

**Fig. 8 ExbBD co-expression is necessary for HasB stabilization.**
Immunodetection of HasB in whole cells of *E. coli* JP313ΔexbBDΔtonB(pHasB33) also harboring pBAD24 (control), pBAD24ExbBD_Ec, pBAD24ExbBD_Sm or pBAD24ExbBD_Ec-Sm76-88, in the presence of either glucose (1 mg/ml) or arabinose (40 µg/ml), indicated by the + signs. The equivalent of $0.2OD_{600nm}$ was loaded in each lane.

of a substrate bound to the extracellular side of a given receptor. In a first model, piston-like movements of TonB drive the unfolding of the plug inside the receptor barrel. In vivo proteolysis studies indicated that the C-terminal domain of TonB can change conformations during the catalytic cycle of protonation and deprotonation of the conserved Asp residue in ExbD TM[40]. In addition, the periplasmic linker between the TM helix and the C-terminal domain of TonB is rich in Pro and Lys residues and might adopt an extended conformation of a sufficient length to span the periplasm[41]. We therefore propose a model in which the force generated through the rotation of HasB/TonB, driven by ExbB rotation around ExbD, is not directly transmitted to the TonB box of the receptor, but could be mediated by the C-terminal domain of ExbD, that might act as an anchor point allowing force transmission and converting the rotation into a pulling force exerted on the TonB/HasB box of the receptor. Further studies are needed to test this model, in particular concerning protein-peptidoglycan interactions, the force needed to trigger TBDT plug opening, and how to distinguish between a rotation and a piston movement. Finally, given the wide range of lag periods we observe in our growth curves using different combinations of ExbB TM1 mutants and TonB/HasB, one may hypothesize that the membrane interaction between ExbB and TonB/HasB influences the rate of transcription activation of the Has locus. In the context of the live bacterium,

where both tonB and hasB are present, the existence of the various potential interactions between ExbBD and TonB or HasB might allow to fine-tune the use of the various iron acquisition systems to the surrounding iron sources. This shows the promiscuity of the ExbBD system from *S. marcescens*, and one might speculate that dedicated ExbBD systems will coexist in a given bacterium with more promiscuous ones.

It is likely that the in vivo entry of a siderophore is a rather slow process: during a cell division time a bacterial cell must take up ca. 300000 iron atoms from its surroundings. Under full induction, there are roughly 10-15000 FepA siderophore receptors per cell and 1500 TonB complexes[32], meaning that each receptor has to undergoes 20 cycles during a generation estimated to 30 min, leading to a turnover time of the TonB complex of about 5–10 s. As compared to flagellar motor that can operate at extremely high speeds (several hundred rotations per second), even though the basic mechanisms are likely to be conserved with the ExbBD/TonB-HasB complex, it is much slower, which likely points to different coupling mechanisms.

## Material and methods

**Strains and plasmid construction**. Strains, plasmids, and oligonucleotides used are shown in Supplementary Table S2. -Plasmid pBADExbBD_Sm was constructed after amplification on the genomic DNA from strain *S. marcescens* Db11 of a ca. 1.42 kb fragment using primers ExbBDSm5' and ExbBDSm3'. The PCR product was purified digested with *Eco*RI and *Sph*I, and ligated with pBAD24 digested with *Eco*RI and *Sph*I. Correct clones were selected after sequencing the insert.

Plasmid pBADExbBDSmHis6 (encoding ExbBSm and a C-terminally His-tagged version of ExbDSm) was constructed by first amplifying on pBADExbBDSm a ca. 0.4 kb fragment with the following oligonucleotides SphIHisCtexbDSm and BglIIexbDSm; after amplification, the fragment was purified, digested with *Bgl*II and *Sph*I, and ligated with pBADExbBDSm digested with the same enzymes. The correct clones were selected after sequencing of the insert. In biological tests, this plasmid was undistinguishable from its parent plasmid pBADExbBDSm.

Plasmid pBADExbBD_Ec was constructed by amplification on genomic MG1655 DNA of a ca. 1.7 kb fragment with the

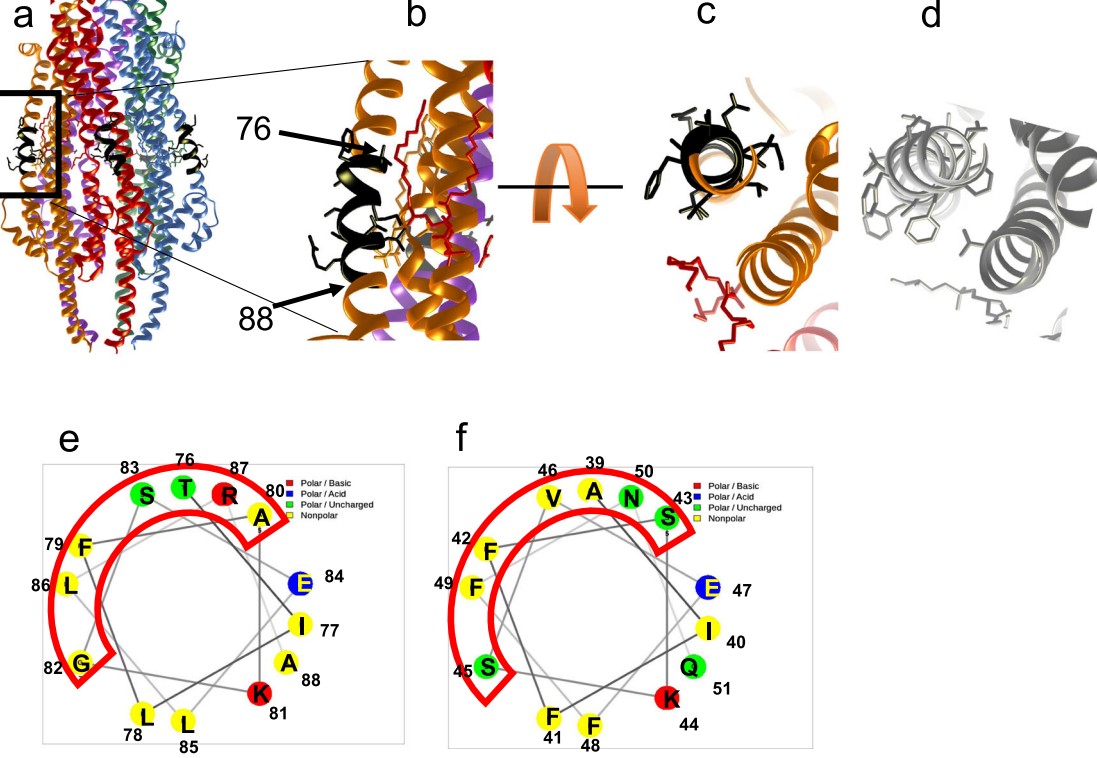

**Fig. 9 ExbB α2 helix: comparison between Sm and Ec.** ExbB-ExbD structure with residues 76-88 colored black and showing the amino acid side chains. **a** general view of the Sm pentamer; (**b**) an enlarged view of (**a**); (**c, d**) two perpendicular close-up views of the exchanged region 76-88 in Sm (**c**) and 39-51 in Ec (**d**). **e, f** Helical wheel representations of the swapped regions between ExbB$_{Sm}$ and ExbB$_{Ec}$ TM1. **e** ExbBSm 76-88; **f**: ExbBEc 39–51. The red boxes show the helical face in interaction with the membrane.

following oligonucleotides ExbBD5c, and ExbBD3c; the PCR product was purified, digested with *Eco*RI and *Sph*I, and ligated with pBAD24 digested with the same enzymes. Correct clones were selected after sequencing.

Plasmid pBADExbB$_{Sm}$His6 (encoding a C-terminally His-tagged version of ExbB$_{Sm}$) was constructed as follows: a PCR fragment was amplified on pBADExbBDSm with the oligos PBADFOR and ExbBHis6, digested with *Eco*RI and *Sph*I, and ligated with pBAD24 digested with the same enzymes. Correct clones were selected by sequencing.

Plasmid pBADExbBD$_{Ec-Sm76-88}$ (encoding a chimeric ExbB$_{Ec}$ protein with its 39–51 residues exchanged for the 76–88 residues from ExbB$_{Sm}$ and ExbD$_{Ec}$) was constructed as follows: Plasmid pBADExbBD$_{Ec}$ was amplified with the two following 5′ phosphorylated oligonucleotides: ExbBEcSm76–88.1 and ExbBEcSm76–88.2. After digestion with *Dpn*I and self-ligation, correct clones were selected by sequencing and the mutated fragments recloned in pBAD24.

Plasmid pBADExbBD$_{Ec-Sm76-84}$ (encoding a chimeric ExbB$_{Ec}$ protein with its 39–47 residues exchanged for the 76–84 residues from ExbB$_{Sm}$ and ExbD$_{Ec}$) was constructed in the same manner, with the following oligonucleotides pairs: ExbBEcSm 76–84.1 and ExbBEcSm 76–84.2.

In the same manner, plasmid pBADExbBD$_{Sm-Ec39-51}$ (encoding a chimeric ExbB$_{Sm}$ protein with its 76–88 residues exchanged for the residues 39–51 from ExbB$_{Ec}$ and ExbD$_{Sm}$) was constructed by PCR on the pBADExbBD$_{Sm}$ plasmid with the following couple of 5′ phosphorylated oligonucleotides: ExbBSmEc39-51.1 and ExbBSmEc39-51.2.

Plasmid pBADExbBD$_{Smdelextss}$ (encoding ExbB$_{Sm}$ deleted of its periplasmic extension and ExbD$_{Sm}$) was constructed similarly, by

using as a template pBADExbBD$_{Sm}$, with the two phosphorylated oligonucleotides ExbBdelextss1 and ExbBdelextss2.

To construct pBADExbB$_{Smdelextss}$His6, a ca. 0.6 kb *Eco*RI-*Kpn*I fragment from pBADExbBD$_{Smdelextss}$ was exchanged for the corresponding fragment of pBADExbBHis6.

Plasmid pAMHasISRADEB (encoding HasI, HasS, HasR, HasA, HasD, HasE and HasB), was constructed by digesting pAMHasISRADE (encoding HasI, HasS, HasR, HasA, HasD and HasE) and pSYC7 (encoding HasD, HasE and HasB)[42] by *Kpn*I and *Hin*dIII, and purifying fragments of respectively ca. 9 and 7 kbases, ligating them together to obtain a plasmid with the whole *has* locus on a low copy number plasmid (pAM238) under the control of its endogenous regulation signals (a fur box upstream of *hasI* and *hasR*, the *hasI* and *hasS* genes respectively encoding the has specific sigma and anti-sigma factors, and the hasS box upstream of *hasS* and *hasR*).

All constructions were carried out in *E. coli* strain XL1-Blue.

Strains and other plasmids used in this study are shown in Supplementary Table S2 and are from the laboratory collection.

**Protein expression and purification.** The BL21DE3 (pExbBD$_{Sm}$His6/pBAD24) or BL21DE3(pExbB$_{Sm}$His6/pBAD24) were grown at 37 °C either in TB or MDM medium, and induced with 0.2% arabinose at 1.5-2OD600nm (TB) or 5-6OD600nm (MDM) and the incubation continued for 3 h. The cells were harvested by centrifugation (10,000 *g* 20 min 4 °C), washed once in 20mMTris-HCl pH8.0, flash-frozen in liquid N₂ and kept at −80 °C. Cells were broken in a Cell disruptor (Constant, UK) at 1kbar (10 g of cells in 40 ml final of 20 mM Tris pH 8.0 containing protease inhibitor (Roche, EDTA free), at 4 °C. Benzonase was

added and after ca. 15 min, the solution was centrifuged for 1 h at 100,000 $g$ at 4 °C. The pellet (crude membrane preparation) was resuspended in 20 mM Tris pH 8.0 plus protease inhibitor cocktail (Roche EDTA free), flash-frozen in liquid $N_2$ and kept at −80 °C.

The crude membrane preparation was solubilized in 20 mM Tris pH 8.0, 20 mM Imidazole, 100 mM NaCl, 10% Glycerol, 0.8% LMNG (10 g of equivalent whole-cell pellet solubilized in 40 ml), plus protease inhibitor cocktail (Roche EDTA-free) for 30 min at 15 °C. After centrifugation (1 h 100,000 $g$), the supernatant was incubated with 2.5 ml Ni-Agarose beads (Thermo-Fisher His-Pure Ni-NTA #88222) preequilibrated with the same buffer except for the detergent concentration (0.0015% LMNG). After 3 h of incubation on a rotating wheel at 4 °C, the beads were washed three times with 25 ml of pre-equilibration buffer and then eluted with two times 25 ml of the pre-equilibration buffer containing 200 mM Imidazole. The eluate was concentrated and washed on 100 kDa cut-off centrifugal device, in pre-equilibration buffer without NaCl and Imidazole. The resulting sample was loaded on a monoQ HR10-100 column equilibrated with 20 mM Tris pH 8.0, 10% glycerol, 0.0015% LMNG and eluted with a gradient from 0 to 1 M NaCl in the same buffer. Peak fractions were collected and concentrated as before. The concentrated sample was then loaded on a Superose 6 increase column equilibrated with 20 mM Tris pH8.0, 100 mM NaCl, 0.0015%LMNG. The peaks fractions were collected, concentrated and their concentration determined by using the theoretical absorption coefficient of either ExbB5D2 (113790 $M^{-1}cm^{-1}$), or ExbB5 (104850$M^{-1}cm^{-1}$). They were then kept frozen at −80 °C in aliquots until use. The SEC profiles as well as a gel of a representative sample after purification are shown in Supplementary Fig. S1. Previous attempts with DDM instead of LMNG yielded similar results, and LMNG was chosen, owing to its very low CMC and its lower background in cryo-EM.

**Activity tests.** Three types of tests were used, either in liquid medium or on agar plates.

1. Growth tests in liquid medium: *E. coli* C600 and its *ΔexbBD* derivatives were transformed with specified plasmids and their growth tested in liquid LB medium at 37 °C, at various dipyridyl concentrations to induce iron starvation. $OD_{600nm}$ was measured after overnight growth.
2. Growth tests on plates: the relevant plasmids were also transformed into *E. coli* C600 *ΔhemA*, a heme auxotroph strain (and derivatives thereof), and growth of the strains was assayed as follows. Briefly, cells were first grown in LB medium (supplemented with delta-aminolevulinic acid (50 μg/ml) to bypass the effect of the *ΔhemA* mutation) at 37 °C up to an $OD_{600nm}$ of 1, and then mixed with melted top agar (0.6%agar in LB), and poured onto LB agar plates containing the appropriate antibiotics with arabinose at a concentration of 40 μg/ml to induce expression of the genes under the $P_{araBAD}$ promoter[43]. Wells, punched with Pasteur pipettes were filled with heme-albumin (at 5, 1 or 0.2 μM), at two dipyridyl concentrations (100 or 200 μM) to induce iron starvation. Plates were incubated overnight at 37 °C and scored for growth around the wells. All experiments were performed in triplicate.
3. Growth curves in liquid medium: a few colonies of *E. coli* C600*ΔhemAΔtonBΔexbBD*(pAMHasISRADEB + pBAD24 or derivatives thereof) or C600*ΔhemAΔexbBD*(pAMHasISRADE + pBAD24 or derivatives thereof) were first inoculated in 4 ml of LB medium at 37 °C with the corresponding antibiotics, and 40 μM or 100 μM dipyridyl respectively, 4 μg/ml arabinose but without delta-aminolevulinic acid. Once the culture reached an $OD_{600nm}$ of ca. 1.2–1.5, it was diluted and inoculated in 48-well Greiner plates, in the same medium to which was added 0.4 μM He-BSA, as a heme

source. The initial $OD_{600nm}$ of the cultures was 0.001. Each well contained 300 μl of growth medium. Duplicates of each strain were made, and the plate was incubated at 37 °C with vigorous shaking (500 rpm) in a Clariostar Plus Microplate reader. $OD_{600nm}$ was recorded every 30 min for 60 h.

**N-terminal sequencing.** The N-terminus of $ExbB_{Sm}$ was determined at the Plateforme Protéomique de l'institut de microbiologie de la Méditerranée (Marseille), after blotting on a PVDF membrane of a purified sample of $ExbBD_{His6}$ run on SDS-PAGE and determined to be APAAN.

**Lipid extraction.** Chloroform (0.20 mL) and methanol (0.40 mL) were sequentially added to a sample of the ExbB or ExbBD complex (0.1 mL). The sample was vortexed for 10 min at room temperature and chloroform (0.2 mL) and water (0.2 mL) were further added. The organic phase was collected and the extraction procedure was repeated on the remaining aqueous phase. Combined organic layers were evaporated to dryness under argon and stored at −20 °C.

**Mass Spectrometry.** Dry lipid extracts obtained from ExbB, the ExbBD complex or from an *E. coli* lysate were solubilized in chloroform/methanol (50/50). Samples were then nano-electrosprayed using a TriVersa NanoMate (Advion Biosciences, Ithaca, USA) coupled to a Synapt G2-Si mass spectrometer (Waters Corporation, Manchester, UK). The instrument was calibrated in negative ion mode (for lipids) and positive ion mode (for proteins) from 50 m/z to 2000 m/z with NaI (50 mg/ml) with an accuracy of 0.8 ppm in resolution mode. The following settings were chosen: sampling cone 40 V, source offset 40 V, source temperature 80 °C, trap gas flow 5 mL/min, helium cell gas flow 180 mL/min. MS/MS spectra were recorded using CID (Collision Induced Dissociation) with a normalized collision energy setup to 30. To measure the mass of the intact proteins of the ExbBD and purified ExbB complexes, a desalting step was performed using micro Bio-Spin™ 6 (BIO-RAD) with 500 mM ammonium acetate. Both samples were analyzed in denaturing conditions after a two-fold dilution with acetonitrile 4% formic acid. Mass spectra were acquired in positive ion mode. All molecular weights were measured after MaxEnt1 software deconvolution into neutral species.

**Cryo-EM grid preparation and data acquisition.** Three micro-liters of either purified $ExbB_{Sm}$ or $ExbBD_{Sm}$ complex at ca. 1 mg/mL was applied to C-Flat 1.2/1.3 holey carbon grids (Protochips Inc., USA) previously glow-discharged in air for 30 s. Grids were blotted for 2 s at blot force 1 and vitrified in liquid ethane using a Vitrobot mark IV (FEI company) operated at 10 °C and 100% relative humidity.

All data collection was performed with a Titan Krios (ThermoFisher Scientific) operated at 300 kV equipped with a K2 Summit direct electron detector (ThermoFisher Scientific) at the European synchrotron research facility, ESRF (Grenoble, France). Movies were recorded in electron counting mode with EPU software (ThermoFisher Scientific), aligned with MotionCor2[44] and aligned images were processed with Gctf[45] using Scipion interface[46].

For ExbB data collection, 4567 movies were collected at a magnification of 165000x with a nominal pixel size of 0.827 Å using a defocus range from −1.5 to −2.5 μm. Movies of 56 frames were acquired using a dose rate of 8 electrons/Å²/s over 7 s, yielding a cumulative exposure of 55.95 electrons/Å².

For ExbBD data collection, 4043 movies were collected at a nominal magnification of ×139,000 with a pixel size of 1.067 Å.

Movies of 48 frames were acquired using a dose rate of 4.6 electrons/Å$^2$/s over 12 s yielding a cumulative exposure of 55.2 electrons/Å$^2$.

## Cryo-EM image processing and analysis

*ExbB*. For ExbB, aligned movies were processed with Gctf[45] and only images with a resolution higher than 4 Å were kept; after visual inspection of the remaining images, processing was carried out with Relion-3[47] (Supplementary Fig. S14). Particles were extracted using a 2-fold binning, issued from a manual picking and a 2-D classification of particles picked out from 50 images. Automatic extraction was performed using the selected 2D class averages. After several rounds of 2D and 3D classification, 161k particles were selected for 3D refinement. They were corrected for local motion using Bayesian polishing option in Relion-3 and a post-refined map produced a 3.1 Å overall resolution with a 5-fold symmetry.

A homology model was built with the *Serratia marcescens* ExbB sequence using Phyre2 server from 5SV0 monomer and docked into the refined map. Refinement was done with Phenix Real Space Refine option with secondary structure, "non-crystallographic symmetry" and Ramachandran restraints[48] and graphically adjusted with Coot[49]. Lipid phosphatidyl glycerol starting structure and geometry were built using Phenix eLBOW[50] and was fit in the map using Coot then refined along with the protein model in Phenix.real_space_refine.

*ExbBD*. The data processing is summarized in Supplementary Fig. S15. Movies were drift-corrected and dose weighted using MotionCorr2[44]. Aligned dose weighted averages were imported into Cryosparc2[51] and contrast transfer function was estimated using CTFFIND4.1[52]. Micrographs with poor CTF estimation statistics or high drift profiles were discarded. The remaining 3028 micrographs were used for automated particle picking. Particles were extracted, Fourier cropped to 2 Å/px and 2D classified. The best 2D classes were used as templates for automated particle picking resulting in 1.3 million particles. After several rounds of classification, the best 600k particles were submitted to 3D classification by means of multi-class ab-initio Reconstruction and Heterogeneous Refinement. 158 k particles belonging to the best-resolved classes were corrected for local motion, re-extracted and used in Non-Uniform Refinement. The resulting refined map has a nominal resolution of 4.56 Å.

Based on the previous map a soft mask lining the micelle was designed in UCSF Chimera[53] and used to signal subtract the corresponding micelle density of particles in the refined map.

Localized refinement of the signal subtracted particles produced a map of the complex with an estimated resolution of 3.96 Å judging by FSC at 0.143 criterion. Data have been deposited both at the EMDB and PDB databases (EMDB 10789 and 11806, PDB 6YE4 and 7AJQ, for ExbB$_{Sm}$ and ExbBD$_{Sm}$ respectively, see Table 1 for refinement statistics).

Unless otherwise specified, the structure figures were made using UCSF Chimera software[53].

## NMR experiments

The C-terminal periplasmic domain of HasB, HasB$_{CTD}$ comprising residues 133–263, was produced and purified as previously reported[28,54]. The peptide corresponding to the periplasmic extension of ExbB$_{sm}$ (A$_1$PAANPAVTESVAPTTA-PAPAAAAPESITPVNPAPTIQPPETRG$_{44}$- numbering with reference to the mature protein) was synthesized by *Proteogenix*.

NMR experiments were acquired at 293 K on a 600 MHz Bruker Avance III spectrometer equipped with a TCI cryoprobe. The spectra were processed with NMRpipe[55] and analyzed with CcpNmr Analysis 2.4 software[56]. Proton chemical shifts were referenced to 2,2-dimethyl-2-silapentane-5 sulfonate as 0 ppm. $^{15}$N were referenced indirectly to DSS[57] (Wishart et al., 1995).

$^1$H–$^{15}$N HSQC experiments were acquired on 0.15 mM HasB$_{CTD}$ in 50 mM sodium phosphate pH 7, 50 mM NaCl with or without the peptide. Aliquots from a solution of peptide at 25 mg/ml in the same buffer were added to the protein sample at 2:1 and 10:1 ratios. Chemical shift perturbations (CSPs) of backbone amide cross-peaks were quantified by using Eq. (1)

$$CSP = [\Delta\delta H^2 + (\Delta\delta N \times 0.15)^2]^{1/2}, \tag{1}$$

where $\Delta\delta H$ and $\Delta\delta N$ are the observed $^1$H and $^{15}$N chemical shift differences between the two experimental conditions.

The $^1$H and $^{15}$N resonance assignments were from Lefevre et al 2007[58] from the biological magnetic resonance data bank (BMRB) #15440 and the structure from the PDB (id 2M2K).

**Other biochemical methods**. SDS-PAGE and immunodetection with anti-HasB or anti-His6 antibodies (Abcam [HIS.H8] (ab18184)) on whole cells or membrane preparations were carried out following standard protocols. Secondary antibodies were coupled to alkaline phosphatase.

**Statistics and reproducibility**. The data pertaining to Figs. 3d and 5a originate from 48-well plate bacterial cultures with a duplicate (Fig. 3d) or quadruplicate (Fig. 5a) measurement. The error bars are standard errors calculated in Microsoft Excel.

**BLAST search**. ExbB$_{Sm}$ was used as a search for BLAST for orthologs in complete bacterial genomes, focusing on "long" ExbB's. Top hits were in *Serratia*, *Yersinia*, *Dickeya*, *Erwinia* and *Pseudomonas*. Those genera were later excluded from successive BLAST searches to obtain orthologs in other genera with higher p-value.

**Reporting summary**. Further information on research design is available in the Nature Research Reporting Summary linked to this article.

## Data availability

Protein structures were deposited into the protein data bank with ID 6YE4 and 7AJQ. Electron microscopy maps were deposited to the EMDB with ID 10789 and 11806.

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

## Acknowledgements

This work was supported by grants from the ANR (HEMESTOCKEXCHANGE ANR-12-BSV3-0022-01 and LABEX DYNAMO ANR-11-LABEX-0011-01). We thank Nathalie Dautin for helpful suggestions and critical reading of the manuscript, as well as Andrew Thompson. We acknowledge Emmanuel Frachon and Christophe Thomas from the Institut Pasteur platform "Production of recombinant proteins" for large scale cultures used in this work, European Synchrotron Radiation Facility for the provision of time on the Titan Krios at beamline CM01, the Institut Pasteur Biological NMR Technological Platform for the use of the NMR spectrometers, the staff at the ESRF CM01 facility in Grenoble for help in the data collection, the staff at the Marseille proteomics facility and Djalila Djouadi and Diana Basto who initially contributed to this project. We thank M. Nilges for interest in this work and B. Miroux for constant support.

## Author contributions

P.D. and V.B. conceived the study and wrote the manuscript. All authors contributed to the manuscript and approved it. P.D. produced protein samples and performed

microbiology experiments. M.C. performed electron microscopy sample screening and data collection. R.J.D.A., P.D.C., V.Y.N.E., and VB processed cryo-EM data. V.B. built structural models. V.B. and P.D. interpreted models. B.L. installed and tested programs. C.M. and J.C.R. performed mass spec analyses. N.I.P. and G.C.A. performed NMR analyses. H.S. provided advice with data collection and processing.

## Competing interests

The authors declare no competing interests.
