## [Peer Review File · Communications Biology]

Reviewers' comments:

Reviewer #1 (Remarks to the Author):

Biou et al. report the cryoEM structures of the ExbB and ExbB-ExbD membrane protein complexes from *Serratia marcescens*. In *S. marcescens*, HasB, a TonB like protein, associates with the ExbBD complex and specifically interacts with the outer membrane transporter HasR to promote heme-iron uptake. The cryoEM structures show a pentameric organization of ExbB, with or without ExbD, defining a transmembrane pore that accommodates two TM domains of ExbD. The ExbBDSm structure is very similar to the previously reported structures of *E. coli* and *P. savastanoi* ExbBD complexes, confirming the 5:2 stoichiometry of this type of motor. The authors identify phospholipids in the ExbB pentameric structure, which they characterize further using mass spectrometry on purified samples. ExbBsm exhibits a N-terminal extension that is commonly found in organisms containing the hasB gene. Using NMR, the authors show that a peptide corresponding to the N-terminal extension of ExbBsm can interact with the HasB C-terminal periplasmic domain. Using mutagenesis experiments, by swapping domains of ExbBsm with ExbBEc, and growth studies in the presence or absence of an iron chelator, the authors claim that the N-terminal extension of ExbBsm, as well as a small domain in the TM1 of ExbBsm, are responsible for the interaction and activity of HasB.

The results reported are interesting and worth publishing in Comm Bio. However extensive revision of the manuscript is needed, including proofreading for English language and clarity, more concise phrasing, better organization, better discussion of the results, and clearer figures. Some information needed to understand the experimental results is not clearly provided, and some figures need to be redone or modified.

For example, the difference between pAMhasISRADEB and pAMhasISRADE is not clearly stated anywhere in the text. In the materials section lines 609-612, it is stated that "pAMhasISRADEB, was constructed by digesting pAMhasISRADE and pSYC7..." but it is not indicated clearly what the addition of "B" stands for.

The growth study shown on figure 3 is difficult to understand, and more contrast is needed to better visualize the halos of cell growth in figure 3-B. Again, without knowing for sure what is the difference between hasISRADEB and hasISRADE, the conclusions from the experiments shown in figure 3-B and stated lines 208-224 are difficult to understand.

What is the position of the his-tag for ExbBsm and ExbBDSm? From the plasmid construction and the mass spectrometry results presented in supp figure 1, this reviewer deduces that the his-tag is on the Cterminus of ExbBsm, and the Cterminus of ExbBDSm for the ExbBDSm complex. Is this correct?

The mass spectrometry experiments performed on organic extraction of ExbBsm and ExbBDSm purified samples allowed to identify phospholipids strongly associated to these complexes. However, these complexes were produced in *E. coli*. Is the lipidic composition of the inner membranes of *S. marcescens* any different from *E. coli*? What is the significance of lipid composition?

The results of the NMR experiments are interesting as it is the first reported interaction between ExbB and the C-terminal globular domain of a TonB-like protein. These results should be discussed in more detail. Are these residues on the Cterm domain of HasB predicted to interact with the TonB/HasB box of HasR? Or are they on the opposite side? These residues should at least be identified by name in the text and labeled in the ribbon representation of figure 5. It would be useful to compare Has vs TonB including their amino acid sequences. Additionally, the legend in figure 5 appears to be written in multiple fonts.

In paragraph 3c are presented the results of cell growth with a ExbBsm mutant lacking the N-terminal extension. Only a lag in the growing curve and a slightly lower maximum OD are observed with this mutant, suggesting that the N-terminal extension of ExbBsm is not required for function. The lag in the growing curve could be due to a lag in the expression of this ExbBsm derivative, as it has been reported that modification of the Nterminal sequence of ExbB can affect expression (ExbBEc with a Nterminal histag does not express for example). What is figure 6-B supposed to show? Figure 6-C shows very faint bands for both ExbBsm and ExbBsmdelextss, and thus would not be visible on the Coomassie stain gel.

The choice of colors for the figure 7-AB and 8 is not optimal, as dark shiny blue does not allow to

see structural details, and the density for the lipids are not clearly visible. There is not a view from the periplasm represented for ExbB_{Sm}. Is the central channel of ExbB_{Sm} open from the periplasm to the cytoplasmic cavity? Or do the periplasmic loops between TM 2 and 3 of ExbB_{Sm} close the channel? Is there any density visible in the hydrophobic channel of ExbB_{Sm} that could be identified as detergent molecules or phospholipids? In figure 11, the choice of tan and pink color is not optimal to highlight the residues 76-88. Staying consistent with the protein coloration would be helpful.

Lines 367-370: the author state that the Asp25 side chains of ExbD face either Thr218 from ExbB chain C, or the interface between ExbB chains A and E. Supp figure 7 shows this conformation; the authors should also show the experimental 3D map of ExbD to assess the fit between the data and the model. What is the local resolution for the ExbD TM domains?

Figure 9: both the tunnels and one of the ExbD chains are colored green, making it difficult to see the ExbD chain. The choice of colors for the ExbB pentamer makes it difficult to interpret the structure as well. It is not indicated in the text what parameters were used to detect these tunnels on the MoleOnline server. Additionally, MoleOnline needs to be cited correctly in line 393 and in the methods. Why do you think the Sm species has this larger channel for transport than *E. coli* or other species?

The results discussed in paragraph 5 are interesting as they potentially identify the region of ExbB interacting with TonB and/or HasB. Such a domain has been described by Deme et al. *Nature Microbiol*, (2020) 5:1553-1564 (extended data fig 10), together with a discussion on the evolutionary co-variation of these residues on ExbB and TonB. The authors should include these data in their discussion.

In the discussion section, lines 509-511, the authors claim that the results of the growth assay with the ExbB_{Sm} "mutant devoid of the periplasmic extension also points to a possible role of the N-terminal periplasmic extension in the activation of the transcription of the has locus". How did the authors reach this conclusion?

The model described in the last sentence of the discussion is rather vague. How would ExbD "act as an anchor point"? To what would it be anchored? the peptidoglycan layer?

Minor comments:

Line 63: "TolQR controls cell division", TolQR does not control cell division, but is involved in the process.

Line 65: "TonB and TolA or" should be "TonB and TolA"

Line 88: what is the mutation in the TM of HasB that restored heme entry? Why is this mutation not discussed further in the text in the light of the results presented?

Line 90: what does "ca" mean? This term is used several times in the text (lines 298, 544, 547, 577, ...)

Line 93: what does a "resident" orthologous of ExbB_{Sm} mean?

Line 106: ref 16 should be 17

Line 115 (and 310): the indicated resolutions 3.2 and 3.9Å are different than the ones reported in Table II.

Line 131: what is a metC homolog and why is it important?

Line 140: all bacterial species should be italicized

Figure 1: in the figure legend, say what the "+" indicates. Additionally, all of the programs with website hyperlinks have specified publications that need to be cited if used in a scientific paper.

This includes Clustral Omega, etc.

Line 222: "a stronger iron starvation is necessary when TonB..." Necessary for what?

Line 224: what is a "genuine" ortholog? What does the "looser specificity" refer to?

Line 266: what protein is being referenced?

Figure 2: properly label axes: ass absorbance to Y-axis and units to the zero value on the X-axis

Figure 4: add units of the X and Y axis. Also, what is the higher band in the ExbB-SEC gel?

Figure 7-D: the TM labels do not show the numbering.

Line 364: "as expected" should be replaced with "as observed for ExbB_{Ec}"

Line 525: This line feels out of place and related to the context of the rest of the paragraph

Lines 551-553: "monomeric TonB devoid of its TM segment was able to drive the partial unfolding of the BtuB plug allowing thus vitamin B12 entry into the periplasm" is misleading. It is merely the interaction between the TonB C-terminal domain and the TonB box of BtuB that was found strong enough to allow the partial unfolding of the BtuB plug domain when a pulling force was applied on TonB.

Lines 561-562: "the C-terminal domain can change conformations...". The C-terminal domain of which protein?

Line 677: "after blotting on a PVDF membrane of a purified sample"

Line 726: how was the initial 3D model determined?

Line 728: the 3.1Å resolution is different than the one reported in Table II

Figure S9: there are no labels in the figure. The general workflow should be explained in more detail. It is impossible to read the bottom labels of the FSC curves. Again, the resolution of 3.1Å is different than the one reported in Table II... The same comments for the FSC curves and resolution apply for figure S10.

For clarity, label all figures with A, B, C, ... instead of right/left and be consistent with figure numbering and the use of "(A)" vs "A."

Reviewer #2 (Remarks to the Author):

The bacterial ExbBExbD complex is associated with TonB/HasB proteins. These complexes harvest energy from the electrochemical proton gradient across the inner membrane to convert it into mechanical energy allowing TonB/HasB to interact with specialized outer membrane transporters for the active import of chelated metals.

In this paper "Structural and molecular determinants for the interaction of ExbB from *S. marcescens* (sm) and HasB", Biou et al. provide a high-resolution cryo-EM structure of ExbBsm oligomer and ExbBExbDsm complex. As for *E. coli* (ec), the structure of ExbBsm corresponds to a pentamer surrounding a central pair of ExbD helices (ExbB5ExbD2). In both structures, the N-terminal extension of ExbB and the periplasmic domain of ExbD are not solved but the C-terminal structure of ExbBsm has been determined. The pentameric pore of ExbBsm adopts a structural plasticity as the size of the pore at the periplasmic side increased upon the presence of the two ExbD transmembrane segments (TM). Interestingly, two channels are clearly detected in the ExbBExbDsm structure while only one is detected in the previously published ExbBExbDec structure. Authors performed the lipid analyses of the complexes. They found that ExbB oligomer associates strongly with specific phospholipids. The EM analysis and mass spectrometry studies have been carried out carefully and fully support the conclusions drawn here. In addition they discovered new determinants for the interaction of ExbBsm with HasB: 1) the N-terminal extension of ExbBsm (extension present in many Alphaproteobacteria and in few other Gammaproteobacteria) with the C-terminal domain of HasB, 2) a sequence of 9 residues in the ExbBsm alpha-helix 2 (membrane-facing residues) with HasB. Overall the findings of this paper are relevant to the field of the structure function of Ton molecular motors. The manuscript should be published after minor revisions addressing the points below.

1 ExbBsm N-terminal extension:

The interaction between the N-terminal extension of ExbBsm and HasB C-terminal domain has been detected by NMR under high concentration of peptides and ask the question of the specificity. A negative control could be added using for example a different synthetic peptide or replacing HasB domain by another protein of a same length. Either a different technique could be checked (plasmon resonance, co-immunoprecipitation).

The residue name of HasB C-terminal domain with a CSP > 0.007 could be indicated on the CSP histogram and on the ribbon structure of HasB (supplementary figure 5 and Figure 5). Because the HSQC spectrum is highly modified upon peptide addition, this interaction may induce conformational changes of HasB. Have the authors considered that all the red residues (Figure 5) interact with ExbB extension? I will expect a short discussion on the residues of HasB with high CSP values, do they differ from those suspected to interact with HasR?

Liquid growth assays with cells producing ExbBsmdelectss and ExbBsm: authors suggest that the ExbBsm extension may activate the transcription of the has locus since a lag is observed only with ExbBsmdelectss. To confirm this suggestion, the liquid growth assay could be performed with cells producing ExbBec-sm76-84 and compare the onset of growth with that of ExbBsmdelectss cells.

2 In the Figure 3, I can detect a faint growth of cells producing ExbBExbDec and HasB only with 200uM DiP? Does this reflect an ExbBec HasB interaction of low affinity? Could liquid growth confirm the absence of growth? This potential interaction will sustain the Discussion on HasB stability (Figure 10, while the suspected stabilization of HasB with ExbDec is very hypothetical since HasB displays low sequence identity with TonB).

3 The functional studies have been well carried out following HasB production levels to confirm the role of the 76-84 sequence of ExbBsm. However, authors had to analyzed the production levels of ExbB variants and HasB to say that « ..ExbBsm-ec39-51 was still active although to a slightly lower extent. »

4 ExbBsm TM1 interaction with HasB

In the discussion section, authors report the first precursor result (Larsen et al 1994) of TonB(deltaV17) by ExbB(A39V). In the cryo-EM results of the ExbBDTonBps complex structure. Deme et al (2020) solved the contour level of a single tube density corresponding to TonB TM surrounding ExbB TM1 and pointed out coevolving residues with potential contact between TonB and ExbB TM1. In the present manuscript authors found a short sequence of non-conserved residues located in the cytoplasmic extension of ExbBsm TM1. The authors could precise these two different regions, first one corresponding to TM1 consensus (Figure 1B) or coevolving (Deme et al, 2020) residue, and second the non conserved (76-86) sequence.

Other comments:

modify reference page 5:in the ExbB pore (17), not (16)

The authors should select a same plasmid name: do hasISRADE/pAM, phasISRADE/238 and pAMhasISRADE, correspond to the same plasmid (id for phasISRADEB)?.

Figure 9: the sizes of the channel entrances may be indicated. I have trouble to visualize the entrance of the ExbBExbDec channel (a lateral view facing the channel would be appropriate)

Figure 11: numbering that appears on the two helical wheels are very small or hidden and could be the same than in the figure legend.

The Figure 11 could be moved in the result section.

Our answers to reviewers' comments are in red

Reviewer #1 (Remarks to the Author):

Biou et al. report the cryoEM structures of the ExbB and ExbB-ExbD membrane protein complexes from *Serratia marcescens*. In *S. marcescens*, HasB, a TonB like protein, associates with the ExbBD complex and specifically interacts with the outer membrane transporter HasR to promote heme-iron uptake. The cryoEM structures show a pentameric organization of ExbB, with or without ExbD, defining a transmembrane pore that accommodates two TM domains of ExbD. The ExbBDSm structure is very similar to the previously reported structures of *E. coli* and *P. savastanoi* ExbBD complexes, confirming the 5:2 stoichiometry of this type of motor. The authors identify phospholipids in the ExbB pentameric structure, which they characterize further using mass spectrometry on purified samples. ExbBsm exhibits a N-terminal extension that is commonly found in organisms containing the hasB gene. Using NMR, the authors show that a peptide corresponding to the N-terminal extension of ExbBsm can interact with the

HasB C-terminal periplasmic domain. Using mutagenesis experiments, by swapping domains of ExbBsm with ExbBEc, and growth studies in the presence or absence of an iron chelator, the authors claim that the N-terminal extension of ExbBsm, as well as a small domain in the TM1 of ExbBsm, are responsible for the interaction and activity of HasB.

The results reported are interesting and worth publishing in Comm Bio. However extensive revision of the manuscript is needed, including proofreading for English language and clarity, more concise phrasing, better organization, better discussion of the results, and clearer figures. Some information needed to understand the experimental results is not clearly provided, and some figures need to be redone or modified.

For example, the difference between pAMhasISRADEB and pAMhasISRADE is not clearly stated anywhere in the text. In the materials section lines 609-612, it is stated that “pAMhasISRADEB, was constructed by digesting pAMhasISRADE and pSYC7...” but it is not indicated clearly what the addition of “B” stands for.

The plasmid nomenclature has been homogenized throughout the manuscript, and a clearer description of the plasmids and what they code for has been included in the Materials and Methods section

The growth study shown on figure 3 is difficult to understand, and more contrast is needed to better visualize the halos of cell growth in figure 3-B. Again, without knowing for sure what is the difference between hasISRADEB and hasISRADE, the conclusions from the experiments shown in figure 3-B and stated lines 208-224 are difficult to understand.

The Figure has been redone and simplified, to show only the most relevant growth conditions. We hope that the contrast is now good enough in this picture. The plate experiment has been redone and pictures taken at two time-intervals, and the equivalent microplate experiment has now been included in Figure 3.

One also must realize that the plates experiments, although quite convenient and nice to look at, are rather crude as they occur in an extremely heterogeneous medium. The bacteria grow on top agar layer, the heme source diffuses from the well, the number of divisions the cells should undergo to obtain a visible halo is quite small as the initial OD_{600nm} is 30mOD compared to 1mOD in the microplate experiment.

What is the position of the his-tag for ExbBSm and ExbBDSm? From the plasmid construction and the mass spectrometry results presented in supp figure 1, this reviewer deduces that the his-tag is on the Cterminus of ExbBSm, and the Cterminus of ExbBDSm for the ExbBDSm complex. Is this correct?

Yes, this is now clearly stated in the text.

The mass spectrometry experiments performed on organic extraction of ExbBSm and ExbBDSm purified samples allowed to identify phospholipids strongly associated to these complexes. However, these complexes were produced in *E. coli*. Is the lipidic composition of the inner membranes of *S. marcescens* any different from *E. coli*? What is the significance of lipid composition?

S. marcescens is a close relative of *E. coli*, and there is no large difference in the phospholipid composition between the two. Two references have been added regarding this point. Yet it appears that ExbBD “selects” a specific set of phospholipids that is not equivalent to the mean composition of the membrane. One might speculate that this might be relevant to the functioning of the complex itself.

The results of the NMR experiments are interesting as it is the first reported interaction between ExbB and the C-terminal globular domain of a TonB-like protein. These results should be discussed in more detail. Are these residues on the Cterm domain of HasB predicted to interact with the TonB/HasB box of HasR? Or are they on the opposite side? These residues should at least be identified by name in the text and labeled in the ribbon representation of figure 5. It would be useful to compare Has vs TonB including their amino acid sequences. Additionally, the legend in figure 5 appears to be written in multiple fonts.

The list of the residues with the highest CSP is shown in the Figure 5 and the Supp Figure 5. They are also indicated and discussed in the text. The legend in Figure 5 was corrected. Line 263: They are mainly located on the helical face of HasB_{CTD} forming a continuous surface of interaction (R175, R178-K180, K184, Q192, T200, L201, Q204, H206, A232, A240, G246). Interestingly, this face is on the opposite side of the third beta strand of HasB that was previously shown interacting with HasR (ref 32). In addition, the residues of a small pocket at the C-terminus of HasB (D255, R259) show also high CSP and can be involved in the interaction with ExbB. The font is now homogeneous.

In paragraph 3c are presented the results of cell growth with a ExbBSm mutant lacking the N-terminal extension. Only a lag in the growing curve and a slightly lower maximum OD are observed with this mutant, suggesting that the N-terminal extension of ExbBSm is not required for function. The lag in the growing curve could be due to a lag in the expression of this ExbBSm derivative, as it has been reported that modification of the Nterminal sequence of ExbB can affect expression (ExbBEc with a Nterminal histag does not express for example).

This lag is reproducible throughout the experiments.

What is figure 6-B supposed to show? Figure 6-C shows very faint bands for both ExbBSm and ExbBSm_{delextss}, and thus would not be visible on the Coomassie stain gel.

As we do not have good antibodies against ExbBSm, we used the his-tagged version to be able to detect the protein in whole cells with anti-His-tag antibodies. The Figure 6-B is just a

loading control, and the Figure 6-C represents the immunodetection of both ExbBSmHis6 and ExbBSmdelexHis6 in whole cells. This is now clearly explained in the text. This experiment alone would not be sufficient to target the function of this extension, but together with the NMR experiment showing interaction between HasBCTD and the extension, it strongly suggests that this extension plays a role either in the uptake process itself, or more likely in the induction of the Has system.

The choice of colors for the figure 7-AB and 8 is not optimal, as dark shiny blue does not allow to see structural details, and the density for the lipids are not clearly visible. There is not a view from the periplasm represented for ExbBSm. Is the central channel of ExbBSm open from the periplasm to the cytoplasmic cavity? Or do the periplasmic loops between TM 2 and 3 of ExbBSm close the channel? Is there any density visible in the hydrophobic channel of ExbBSm that could be identified as detergent molecules or phospholipids? In figure 11, the choice of tan and pink color is not optimal to highlight the residues 76-88. Staying consistent with the protein coloration would be helpful.

we changed the coloring for the monomers so that they are now more contrasted and homogeneous throughout the paper. The central channel is open from both sides but cryo-EM density observed at low level shows density in the channel. This density is disordered and did not allow us to distinguish between detergent and phospholipid. Supplementary figure S7 and comments in the text have been added to document this observation.

Lines 367-370: the author state that the Asp25 side chains of ExbD face either Thr218 from ExbB chain C, or the interface between ExbB chains A and E. Supp figure 7 shows this conformation; the authors should also show the experimental 3D map of ExbD to assess the fit between the data and the model. What is the local resolution for the ExbD TM domains? Figure 9: both the tunnels and one of the ExbD chains are colored green, making it difficult to see the ExbD chain. The choice of colors for the ExbB pentamer makes it difficult to interpret the structure as well. It is not indicated in the text what parameters were used to detect these tunnels on the MoleOnline server. Additionally, MoleOnline needs to be cited correctly in line 393 and in the methods. Why do you think the Sm species has this larger channel for transport than *E. coli* or other species?

the experimental map was added to Figure S7. Figure 9 was redrawn with colors consistent with the rest of the paper. We do not know why Sm has a larger channel than Ec. One may invoke the different experimental setups, as the cryo-EM for *E. coli* was carried out in nanodiscs, whereas our study was done in detergent.

The results discussed in paragraph 5 are interesting as they potentially identify the region of ExbB interacting with TonB and/or HasB. Such a domain has been described by Deme et al. *Nature Microbiol*, (2020) 5:1553-1564 (extended data fig 10), together with a discussion on the evolutionary co-variation of these residues on ExbB and TonB. The authors should include these data in their discussion.

The reference has been included, together with a short discussion

In the discussion section, lines 509-511, the authors claim that the results of the growth assay with the ExbBSm “mutant devoid of the periplasmic extension also points to a possible role of

the N-terminal periplasmic extension in the activation of the transcription of the has locus”. How did the authors reach this conclusion?

It is not yet a conclusion, it is only a suggestion for the time being. However, as in our experimental setting the growth is the result of both the transcription activation through the HasI/HasS couple, and that ExbBSm and ExbBSmdelex, seem to be expressed in the same amounts, it is a reasonable assumption that transcription activation might be the functional target of the periplasmic extension of ExbBSm. It is well beyond the scope of this study to bring evidence for this hypothesis.

The model described in the last sentence of the discussion is rather vague. How would ExbD “act as an anchor point”? To what would it be anchored? the peptidoglycan layer?

It is the most obvious possibility, and anchoring to the peptidoglycan (although as yet without experimental support) has been previously suggested by JC Deme et al. (ref 17)

Minor comments:

Line 63: “TolQR controls cell division”, TolQR does not control cell division, but is involved in the process.

Corrected.

Line 65: “TonB and TolA or” should be “TonB and TolA”

Corrected.

Line 88: what is the mutation in the TM of HasB that restored heme entry? Why is this mutation not discussed further in the text in the light of the results presented?

A description of the hasB6 mutant has been added and, in this study, we did not want to address the HasB/TonB vs. ExbBD specificity from the HasB/TonB point of view.

Line 90: what does “ca” mean? This term is used several times in the text (lines 298, 544, 547, 577, ...)

An abbreviation for circa, now explained.

Line 93: what does a “resident” orthologous of ExbBDSm mean?

It was just meant to refer to *Serratia*, we suppressed it, as it was superfluous.

Line 106: ref 16 should be 17

Corrected.

Line 115 (and 310): the indicated resolutions 3.2 and 3.9Å are different than the ones reported in Table II.

the resolutions have been changed to those shown on FSC curves from Supplementary figures S10 and S11: 3.1 and 3.96Å.

Line 131: what is a metC homolog and why is it important?

This just deals with the gene environment of a given locus across species (synteny). In this case, genetic organization is conserved on the 3' side, but not on the 5' side. The *metC* gene itself has no specific relevance.

Line 140: all bacterial species should be italicized

Corrected.

Figure 1: in the figure legend, say what the "+" indicates. Additionally, all of the programs with website hyperlinks have specified publications that need to be cited if used in a scientific paper. This includes Clustral Omega, etc.

Corrected.

Line 222: "a stronger iron starvation is necessary when TonB..." Necessary for what?

This has been corrected for: "higher Dipyridyl concentration (corresponding to a stronger iron starvation) is required for heme entry in the reconstituted system..."

Line 224: what is a "genuine" ortholog? What does the "looser specificity" refer to?

The phrase has been modified to: "Therefore, ExbBD_{Sm} is the *E. coli* ExbBD ortholog, able to associate both with HasB and TonB_{Ec}"

Line 266: what protein is being referenced?

ExbB, this is now indicated.

Figure 2: properly label axes: ass absorbance to Y-axis and units to the zero value on the X-axis

Corrected.

Figure 4: add units of the X and Y axis. Also, what is the higher band in the ExbB-SEC gel?

Done. The higher band on the gel has been identified by mass spec as AcrB.

Figure 7-D: the TM labels do not show the numbering.

labels have been added and Figure 7D is now 7C.

Line 364: "as expected" should be replaced with "as observed for ExbB_{Ec}"

Corrected.

Line 525: This line feels out of place and related to the context of the rest of the paragraph

We did not really understand this point raised by the reviewer

Lines 551-553: “monomeric TonB devoid of its TM segment was able to drive the partial unfolding of the BtuB plug allowing thus vitamin B12 entry into the periplasm” is misleading. It is merely the interaction between the TonB C-terminal domain and the TonB box of BtuB that was found strong enough to allow the partial unfolding of the BtuB plug domain when a pulling force was applied on TonB.

Corrected.

Lines 561-562: “the C-terminal domain can change conformations...”. The C-terminal domain of which protein?

Corrected.

Line 677: “after blotting on a PVDF membrane of a purified sample”

Corrected.

Line 726: how was the initial 3D model determined?

as stated in the text:

A homology model was built with the *Serratia marcescens* ExbB sequence using Phyre2 server from 5SV0 monomer and docked into the refined map. Refinement was done with Phenix Real Space Refine option with secondary structure, “non-crystallographic symmetry” and Ramachandran restraints (46) and graphically adjusted with Coot (26). Lipid phosphatidyl glycerol starting structure and geometry were built using Phenix eLBOW (47) and was fit in the map using Coot then refined along with the protein model in Phenix.real_space_refine.

Line 728: the 3.1Å resolution is different than the one reported in Table II

this was corrected.

Figure S9: there are no labels in the figure. The general workflow should be explained in more detail. It is impossible to read the bottom labels of the FSC curves. Again, the resolution of 3.1Å is different than the one reported in Table II... The same comments for the FSC curves and resolution apply for figure S10.

more details were added to Fig S9 and resolution figures were corrected.

For clarity, label all figures with A, B, C, ... instead of right/left and be consistent with figure numbering and the use of “(A)” vs “A.”

Corrected.

Reviewer #2 (Remarks to the Author):

The bacterial ExbBExbD complex is associated with TonB/HasB proteins. These complexes harvest energy from the electrochemical proton gradient across the inner membrane to convert it into mechanical energy allowing TonB/HasB to interact with specialized outer membrane transporters for the active import of chelated metals.

In this paper “Structural and molecular determinants for the interaction of ExbB from *S. marcescens* (sm) and HasB”, Biou et al. provide a high-resolution cryo-EM structure of ExbBsm oligomer and ExbBExbDsm complex. As for *E. coli* (ec), the structure of ExbBsm corresponds to a pentamer surrounding a central pair of ExbD helices (ExbB5ExbD2). In both structures, the N-terminal extension of ExbB and the periplasmic domain of ExbD are not solved but the C-terminal structure of ExbBsm has been determined. The pentameric pore of ExbBsm adopts a structural plasticity as the size of the pore at the periplasmic side increased upon the presence of the two ExbD transmembrane segments (TM). Interestingly, two channels are clearly detected in the ExbBExbDsm structure while only one is detected in the previously published ExbBExbDec structure. Authors performed the lipid analyses of the complexes. They found that ExbB oligomer associates strongly with specific phospholipids. The EM analysis and mass spectrometry studies have been carried out carefully and fully support the conclusions drawn here. In addition they discovered new determinants for the interaction of ExbBsm with HasB: 1) the N-terminal extension of ExbBsm (extension present in many Alphaproteobacteria and in few other Gammaproteobacteria) with the C-terminal domain of HasB, 2) a sequence of 9 residues in the ExbBsm alpha-helix 2 (membrane-facing residues) with HasB

Overall the findings of this paper are relevant to the field of the structure function of Ton molecular motors. The manuscript should be published after minor revisions addressing the points below.

1 ExbBsm N-terminal extension:

The interaction between the N-terminal extension of ExbBsm and HasB C-terminal domain has been detected by NMR under high concentration of peptides and ask the question of the specificity. A negative control could be added using for example a different synthetic peptide or replacing HasB domain by another protein of a same length. Either a different technique could be checked (plasmon resonance, co-immunoprecipitation).

The Interaction between a protein and a peptide is usually with a low affinity. As asked by Reviewer 2, we show here an NMR experiment with HasB_{CTD} and a different synthetic peptide (L17T) with the same range of protein and peptide concentration as that was used in the manuscript.

No chemical shift perturbations of HasB_{CTD} are observed even with a high amount of peptide. The L17T and ExbB_{sm} peptides are comparable in terms of composition, with positively and negatively charged residues as well as hydrophobic ones.

The residue name of HasB C-terminal domain with a CSP > 0.007 could be indicated on the CSP histogram and on the ribbon structure of HasB (supplementary figure 5 and Figure 5). Because the HSQC spectrum is highly modified upon peptide addition, this interaction may induce conformational changes of HasB. Have the authors considered that all the red residues (Figure 5) interact with ExbB extension?

The figure 5 and the supplementary figure 5 have been modified as it was also suggested by Reviewer 1. The interacting residues are discussed in the text. All the residues displaying a CSP > 0.007 are considered for the analysis except the last residues.

I will expect a short discussion on the residues of HasB with high CSP values, do they differ from those suspected to interact with HasR?

Line 263: They are mainly located on the helical face of HasB_{CTD} forming a continuous surface of interaction (R175, R178-K180, K184, Q192, T200, L201, Q204, H206, A232, A240, G246). Interestingly, this face is on the opposite side of the third beta strand of HasB that was previously shown interacting with HasR (ref 32). In addition, the residues of a small pocket at the C-terminus of HasB (D255, R259) show also high CSP and can be involved in the interaction with ExbB.

Liquid growth assays with cells producing ExbBsmdelectss and ExbBsm: authors suggest that the ExbBsm extension may activate the transcription of the has locus since a lag is observed only with ExbBsmdelectss. To confirm this suggestion, the liquid growth assay could be

performed with cells producing ExbBec-sm76-84 and compare the onset of growth with that of ExbBsmdelectss cells.

This was an excellent suggestion by the reviewer, that we did not initially intend to include in this study. This experiment is now shown Figure 6 and the onset of growth of this chimeric protein occurs quite earlier than that of ExbBDEc, but not as quickly as with ExbBsmdelect.

2 In the Figure 3, I can detect a faint growth of cells producing ExbBExbDec and HasB only with 200uM DiP? Does this reflect an ExbBec HasB interaction of low affinity? Could liquid growth confirm the absence of growth? This potential interaction will sustain the Discussion on HasB stability (Figure 10, while the suspected stabilization of HasB with ExbDec is very hypothetical since HasB displays low sequence identity with TonB).

The reviewer is right, and this Figure has been modified (it is a question of kinetics cf. comment to previous reviewer), and we have extended the discussion on the stabilization of HasB by ExbBDEc. We are presently investigating whether the complete absence of growth in the previous study (ref 12), could actually be due to different genetic background for the strains used: in fact, one of the two *tonB* mutants had a large deletion of a chromosomal region encompassing in particular the *clsA* gene that encodes the main cardiolipin synthase during exponential growth phase. The other strain had an uncharacterized *tonB* mutation which we have now deciphered: the *tonB* gene is interrupted at codon 102 by an insertion sequence giving rise to a chimeric protein of 110 residues that might interfere with normal TonB/HasB function. We feel that any conclusion from this work should be the subject of another study.

3 The functional studies have been well carried out following HasB production levels to confirm the role of the 76-84 sequence of ExbBsm. However, authors had to analyzed the production levels of ExbB variants and HasB to say that « ..ExbBsm-ec39-51 was still active although to a slightly lower extent. »

Although we do not have the formal proof that this protein is made at similar levels as the ExbBsm protein, this is quite likely, but we have modified the sentence to account for the reviewer's point.

4 ExbBsm TM1 interaction with HasB

In the discussion section, authors report the first precursor result (Larsen et al 1994) of TonB(deltaV17) by ExbB(A39V). In the cryo-EM results of the ExbBDTonBps complex structure. Deme et al (2020) solved the contour level of a single tube density corresponding to TonB TM surrounding ExbB TM1 and pointed out coevolving residues with potential contact between TonB and ExbB TM1. In the present manuscript authors found a short sequence of non-conserved residues located in the cytoplasmic extension of ExbBsm TM1. The authors could precise these two different regions, first one corresponding to TM1 consensus (Figure 1B) or coevolving (Deme et al, 2020) residue, and second the non conserved (76-86) sequence.

See answer to reviewer 1.

Other comments:

modify reference page 5:in the ExbB pore (17), not (16)

done

The authors should select a same plasmid name: do hasISRADE/pAM, phasISRADE/238 and pAMhasISRADE, correspond to the same plasmid (id for phasISRADEB)?.

See answer to reviewer 1

Figure 9: the sizes of the channel entrances may be indicated. I have trouble to visualize the entrance of the ExbBExbDec channel (a lateral view facing the channel would be appropriate)

The channel diameters were added to the figure legend: 3Å for Sm and 2Å for Ec. We now hope that the Figure with changed colors is easier to visualize.

Figure 11: numbering that appears on the two helical wheels are very small or hidden and could be the same than in the figure legend.

changed.

The Figure 11 could be moved in the result section.

this was done.

REVIEWERS' COMMENTS:

Reviewer #1 (Remarks to the Author):

The authors have addressed all of my comments satisfactorily.

Reviewer #2 (Remarks to the Author):

Authors greatly improved their manuscript, the reading is now easy with a clearer plasmid description, NMR results, and discussion on TonB/HasB interactions while some figures have been upgraded according to the reviewers' comments. Specifically, new experiments have been carried out to give the new Figure 3B-C-D which demonstrated that ExbBDec is functional with HasB (adding the discussion on HasB stabilization by ExbBDec). New experiments have been added in the Figure 6A that clearly indicate the role of the ExbBsm N-terminal extension to activate the transcription of the has genes.

Just a little comment, the yellow/orange colored line in the bottom part of the Figure 6C could be removed.